# SHIFT: A DEM-Based Spatial Heterogeneity Improved Mapping of Global Geomorphic Floodplains

Kaihao Zheng[1], Peirong Lin[1, 2, *], Ziyun Yin[1]

1. Institute of Remote Sensing and GIS, School of Earth and Space Sciences, Peking University, Beijing, China.

2. International Research Center of Big Data for Sustainable Development Goals, Beijing, China.

* Corresponding author: Peirong Lin (peironglinlin@pku.edu.cn)

Manuscript submitted to *ESSD*, December 29[th], 2023

Revised manuscript submitted to *ESSD*, June 16[th], 2024

## Abstract

Floodplain is a vital part of the global riverine system. Among all the global floodplain delineation strategies empowered by remote sensing, DEM-based delineation is considered computationally efficient with relatively low uncertainties, but the parsimonious model struggles with incorporating basin-level spatial heterogeneity of the hydrological and geomorphic influences into the map. In this study, we propose a globally applicable thresholding scheme for DEM-based floodplain delineation to improve the representation of spatial heterogeneity. Specifically, we develop a stepwise approach to estimate the Floodplain Hydraulic Geometry (FHG) scaling parameters for river basins worldwide at the scale of Level-3 HydroBASINS to best respect the scaling law while approximating the spatial extent of two publicly available global flood maps derived from hydrodynamic modeling. The estimated FHG exponent exhibits a significant positive relationship with the basins' hydroclimatic conditions, particularly in 33 of the world's major river basins, indicating the ability of the approach to capture fingerprints from heterogeneous hydrological and geomorphic influences. Based on the spatially-varying FHG parameters, a ~90-m resolution global floodplain map named Spatial Heterogeneity Improved Floodplain by Terrain analysis (SHIFT) is delineated, which takes the hydrologically corrected MERIT-Hydro dataset as the DEM inputs and the Height Above Nearest Drainage (HAND) as the terrain attribute. Our results demonstrate that SHIFT validates better with reference maps than both hydrodynamic modeling and DEM-based

approaches with universal parameters. The improved delineation is mainly with better differentiation between mainstreams and tributaries in major basins and a more comprehensive representation of stream networks in aggregated river basins. SHIFT estimates global floodplain area to be 9.91 million km$^2$, representing 6.6% of the world's total land area. SHIFT data layers are available at two spatial resolutions (90-m and 1-km), along with the updated parameters, at

https://zenodo.org/records/11835133 (Zheng et al., 2024). We anticipate SHIFT to be used to support applications requiring boundary delineations of the global geomorphic floodplains.

## Highlights

- We develop a globally applicable thresholding scheme for DEM-based floodplain mapping
that improves the integration of floodplain spatial heterogeneity.
- We create a new 90-m geomorphic floodplain map named Spatial Heterogeneity Improved Floodplain by Terrain analysis (SHIFT).
- SHIFT has better delineation of mainstreams in major river basins and more comprehensive representation of stream networks in aggregated river basins.
- The estimated exponent in Floodplain Hydraulic Geometry (FHG) exhibits a statistically significant positive relation with hydroclimatic factors.
- Global floodplain area is estimated to be 9.91 million km$^2$, representing 6.6% of the world's total land area.

# 1. Introduction


Floodplain is an integral component of the global riverine system – it acts as a river's ecological buffer, and offers conveniences for human settlements while also harboring flood risks (Di Baldassarre et al., 2013). Floodplains accommodate over half of the world's human habitation and development due to their favorable nature (Andreadis et al., 2022; Best, 2019). Thus, accurate

delineation of floodplain boundaries has attracted wide attention among ecologists, flood practitioners/engineers, and geomorphologists (Wohl, 2021). Among various mapping efforts across different scales and resolutions (Dhote et al., 2023), global-scale floodplain maps are particularly valuable as they require a consistent and spatially continuous framework, which can be leveraged to offer insights into the changing global floodplain characteristics and flood risks

(Du et al., 2018; Lindersson et al., 2020; Rajib et al., 2021, 2023; Rentschler et al., 2022, 2023).

Terrestrial observation empowered by satellite remote sensing provides essential data that allow for the delineation of global-scale floodplain by estimating inundation caused by flood extremes. One strategy for the delineation is to directly detect the flood inundation areas from optical or

Synthetic Aperture Radar (SAR) remote sensing imageries (e.g., Tellman et al., 2021). This requires historical occurrence of a flood event to define a floodplain, but such an event-based approach often results in spatially discrete global floodplain maps limited by satellite data quality and accessibility. It also overlooks unflooded yet at-risk locations, potentially underestimating floodplain extents. Other strategies involve running hydrodynamic or hydraulic models, which

takes input data from terrain and runoff forcing and then simulate detailed flood inundation dynamics in a computationally demanding manner (Bates et al., 2018; Trigg et al., 2021). This method derives continuous floodplain maps, and it emphasizes the inundation area under different flood return periods (e.g., 100-year floodplain), which is more commonly used in engineering and hazard mitigation practices (Wohl, 2021). Various global floodplain maps are available from

different hydrodynamic models, including the European Commission's Joint Research Centre (JRC) (Dottori et al., 2016), the CIMA-UNEP model from the Global Assessment Report (GAR) (Rudari et al., 2015), CaMa-Flood (Yamazaki, 2014), Fathom Global (Sampson et al., 2015) and GLOFRIS (Winsemius et al., 2013). Yet due to the uncertainties that concern the forcing inputs, model structure and parameters, notable inconsistencies are reported across these datasets (Bates,

2023; Bernhofen et al., 2022; Trigg et al., 2016). Thus, the uncertainties associated with the above

approaches highlight the need for continuous efforts to improve global floodplain mapping strategies.

Recent advancements in remote sensing offer ever-growing spatial coverage, refined resolution and improved accuracy of global terrain products, motivating the third strategy to directly delineate floodplains with satellite-derived terrain data. The Digital Elevation Models (DEMs)-based or terrain analysis approach is often considered to exhibit higher computational efficiencies as it takes less data and parameters, and the sufficiently accurate DEMs are already recognized as the least uncertain component compared to other uncertainty sources in global floodplain mapping with hydrodynamic models (Bates, 2023). As a result, the parsimonious DEM-based floodplain mapping method receives growing attention in large-scale studies and ungauged basins (Manfreda et al., 2014; Nardi et al., 2013, 2018; Tavares da Costa et al., 2019). DEM-based floodplain mapping generally consists of two steps. First, essential terrain attributes such as Height Above Nearest Drainage (i.e., HAND), Topography Wetness Index, Slope Position, or their derivatives are calculated from DEMs to represent river proximity (Beven and Kirkby, 1979; Rennó et al., 2008; Weiss, 2001; Xiong et al., 2022). Second, thresholding schemes are applied to these attributes to delineate the floodplain boundary (Dhote et al., 2023). For example, the GFPlain algorithm, a widely applied method for terrain-based floodplain delineation (Knox et al., 2022; Manfreda et al., 2014; Nardi et al., 2006; Rajib et al., 2023), adopts such an approach to create the GFPlain250m dataset (Nardi et al., 2019). In a recent comparative study, GFPlain250m was proved to show the highest consistency with several existing floodplain maps, highlighting the potential of geomorphic floodplain delineation in reducing model uncertainties (Lindersson et al., 2021).

However, DEM-based mapping methods also face challenges particularly in characterizing spatial heterogeneity (Annis et al., 2019), or spatial variations of floodplain characteristics and processes discovered across scales such as topography, morphology, climate, stratigraphy, biodiversity and river fluxes (Iskin and Wohl, 2023; Wohl, 2021; Wohl and Iskin, 2019). In a DEM-based mapping approach, one generally addresses the impact of heterogeneous factors on floodplain extents through thresholding schemes, but currently there is no universal large-scale thresholding scheme available (Dhote et al., 2023). Many previous attempts assume homogeneous determining factors within the study area and directly assume a universal threshold (e.g., a specific HAND threshold

for all pixels) in obtaining geomorphic floodplains, which may suffice at smaller scales but could significantly skew results in large-scale studies (Afshari et al., 2018; Hocini et al., 2021; Manfreda et al., 2014; Nardi et al., 2013). To better account for spatial heterogeneity, the aforementioned

GFPlain algorithm (Nardi et al., 2006) applied the floodplain hydraulic geometry (hereafter FHG, Bhowmik, 1984) as the foundation of their thresholding scheme. In the FHG scaling law relationship, the floodplain extent scales exponentially with the river's upstream drainage area (UPA), which adds UPA as the primary determining factor in deriving floodplain maps. However, it often adopts universal values for FHG parameters across basins (Nardi et al., 2019), implying

that other sources of heterogeneity encapsulated by the FHG scaling parameters is ignored. While studies attempting to estimate the empirical parameters of FHG with statistical fitting methods exist, it remains difficult to derive FHG parameters worldwide and to offer further physical interpretations for the parameters. Such inadequate representation and understanding of spatial heterogeneity in FHG parameters may lead to inaccurate delineations in less well-documented

regions; for example, overestimated floodplains in arid or semi-arid area as reported by existing assessments of geomorphic floodplains (Lindersson et al., 2021).

To complement existing studies, here we develop a globally applicable framework to estimate FHG parameters that better integrate spatial heterogeneity into our thresholding scheme. It takes

two publicly available hydrodynamic floodplain maps as the reference to estimate spatially varied FHG parameters across all global river basins at the scale of Level-3 HydroBASINS. Based on this, we develop a 90-m global geomorphic floodplain map named Spatial Heterogeneity Improved Floodplain by Terrain analysis (SHIFT). SHIFT calculates HAND above the nearest river pixel to which it drains by utilizing the hydrologically corrected MERIT-Hydro (Yamazaki et al., 2019)

dataset. Due to the use of the MERIT-Hydro dataset, due to which SHIFT also addresses limitations of existing global geomorphic mapping that used uncorrected DTM with limited spatial coverages (60°N to 60°S) and relatively low spatial resolutions. Our manuscript is organized as the follows. Section 2 introduces our methods and data in detail. Section 3 presents our geomorphic floodplain data and the accuracy assessment against several reference maps. Sections 4 & 5 close with

discussions and conclusions of this study.

# 2. Methods and Data

SHIFT is developed following the technical flowchart in **Fig. 1**. Below we will describe our data and methods in detail.

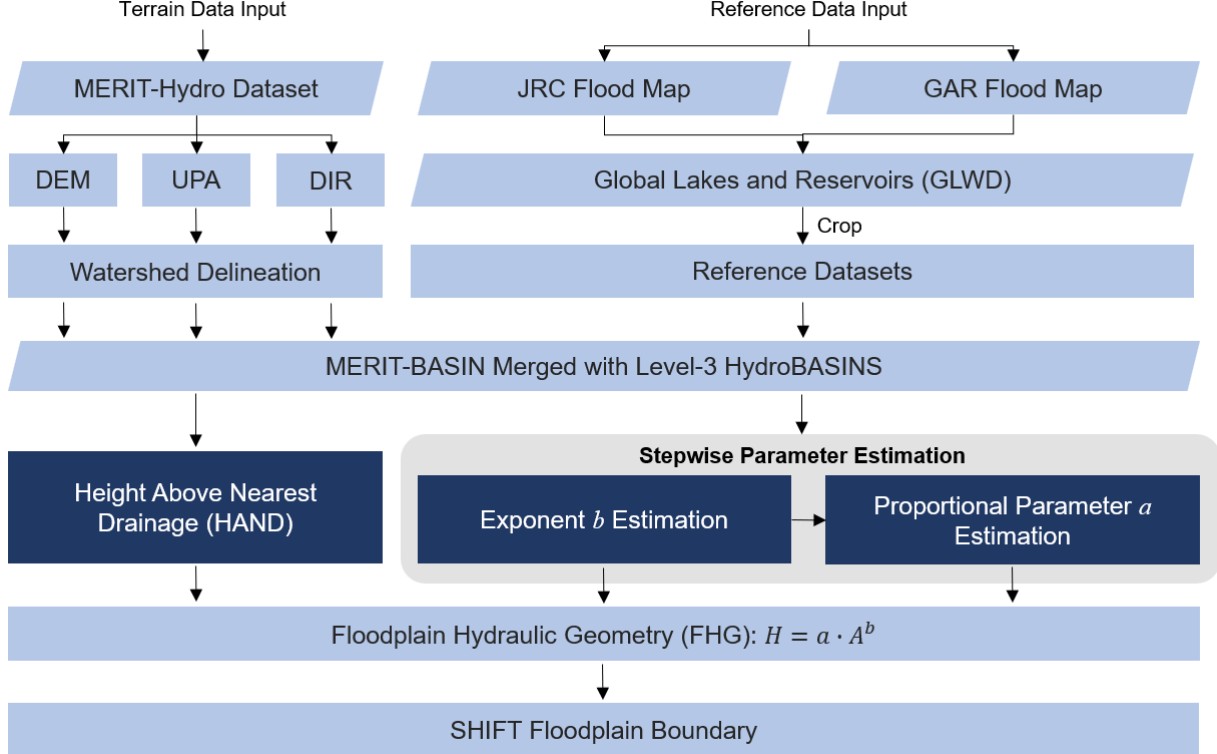

**Figure 1. Technical workflow of the study.** Parallelograms denote data; rectangles denote processing; highlighted rectangles are the key features of SHIFT. Stepwise parameter estimation is marked in the grey box.


## 2.1 Data

### 2.1.1 Terrain Data

*(1) MERIT-Hydro hydrography map.* We take terrain inputs (i.e., elevation, d-8 flow direction and upstream drainage area) from the MERIT-Hydro dataset (Yamazaki et al., 2019). It is a 90-m resolution global dataset that combines data from the Space Shuttle Radar Topography Mission (SRTM) and airborne LiDAR, which has undergone rigorous error correction processes to remove various types of errors such as striping noise, speckle noise, absolute errors, as well as biases in

tree heights. Multiple remote sensing datasets and Volunteer Geographic Information System (Volunteer GIS) water data are used to further enhance its ability to identify river locations. Specifically, it combines OpenStreetMap river vector data, SRTM waterbody data, and Landsat-derived water data to calculate the likelihood of a grid cell representing a water body. In areas with a high likelihood of water, the elevation is adjusted lower. This approach effectively improves the accuracy of flow direction calculations and minimizes deviations in flat areas. The dataset is georeferenced to the WGS84 and EGM96 geodetic reference systems, with a spatial resolution of 3 arc-seconds (approximately 90 meters at the equator).

*(2) HydroBASINS Global Basins.* We applied basin boundary data from the Level-3 HydroBASINS dataset to introduce the basin-by-basin spatial variability in parameter estimation. It is a multi-level global basin dataset derived from the SRTM DEM data as part of the HydroSHEDS project (Lehner and Grill, 2013). HydroBASINS is structured into 12 levels of basins, with higher levels representing finer basins. The dataset applies the Pfafstetter coding system to support analysis of watershed topology including upstream and downstream connectivity. The first three levels are assigned with Leve-1 categorizing continents, Level-2 dividing continents into major sub-units, and Level 3 delineating the largest river basins on each continent (Lehner, 2014). The Level-3 sub-basins in HydroBASINS consist of 269 units globally with an average size of 555,600 km$^2$. Level-4 and Level-5 boundaries are also applied for further analysis on scales.

*(3) MERIT-Basins.* MERIT-Basins is a global vector hydrography database derived from the 90-m MERIT-Hydro product, based on a 25 km² threshold for drainage areas (Lin et al., 2019). It aligns well with the MERIT-Hydro dataset. To obtain the corresponding boundaries for parameter estimation, we combined MERIT-Basins into groups equivalent to Level-3 to Level-5 of HydroBASINS. We aggregated MERIT-Basins based on its spatial relationship with basins from HydroBASINS, ensuring that the centroid of a MERIT-Basin falls within the corresponding boundary. This approach accounts for slight differences in boundaries due to the use of different terrain data, preventing confusion in hydrological representation. Among the Level-3 basins, the 40 largest hydrologically connected basins were manually selected, based on the hypothesis that connected basins better apply the scaling law due to shared attributes within the same hydrological

system. Seven of these 40 basins, with centroid located above 60°N, were excluded since one of our reference maps does not cover regions above 60°N.

190

### 2.1.2 Reference and benchmark datasets

*(1) JRC flood map.* The flood hazard map created by the European Commission's JRC is selected as part of the reference and validation dataset. It is based on the 3-arcsecond SRTM DEM, which combines hydrological simulations from the Global Flood Awareness System (GloFAS) with a two-dimensional CA2D hydraulic model for flood inundation mapping. The GloFAS simulations utilize ERA-Interim data, covering the period from 1980 to 2013, and operate at a resolution of 0.1 degrees (approximately 11 km at the Equator). The system simulates streamflow by coupling two distributed global models: HTESSEL, which estimates surface water and energy fluxes in response to atmospheric forcing, and LISFLOOD Global, which uses the output from HTESSEL to simulate routing processes and streamflow. The flood hazard maps produced are at a 30-second resolution and focus on river channels with an upstream catchment area greater than 5000 km² (Dottori et al., 2016). The JRC dataset provides flood hazard maps with different return periods from 10 years to 1000 years. Here we used the 500-year flood map as a reference floodplain map based on the notion that geomorphic floodplains are dominantly shaped by high-impact yet low-possibility events (Annis et al., 2019; Bhowmik, 1984; Lindersson et al., 2021).

*(2) GAR flood map.* We also select the 500-year flood map by hydrodynamic model from the GAR of the United Nations Office for Disaster Risk Reduction (UNDRR) and the CIMA foundation as a reference and validation dataset. The GAR data employs a global database of discharge data from over 8000 stations to estimate extreme streamflows and DEM from HydroSHEDS for hydraulic modeling. This one-dimensional model applies Manning's equation to calculate river stages. The GAR flood map also considers artificial flood defense by assuming target return periods of flood defenses based on the GDP distribution, thereby locally reducing the estimated flooded volume within the estimated protected area. The dataset is characterized by return periods of 25, 50, 100, 200, 500, and 1000 years, a coverage of 60°N to 60°S, with a native resolution of 90 meters from the SRTM DEM, later aggregated to 1-km for risk computation (Rudari et al., 2015).

*(3) GFPlain250m floodplain map.* The aforementioned geomorphically delineated GFPlain250m floodplain map is used as the benchmark and another validation dataset (Nardi et al., 2019). It shares the same coverage with GAR (60°N to 60°S). For each grid, it calculates the height above the lowest elevation grid within the same watershed (i.e., the basin outlet) as the terrain attribute rather than the nearest river grid to which it drains. This exaggerates the vertical distance to streams for upstream pixels and may thus lead to underestimation of floodplain. FHG is applied as the thresholding scheme (Nardi et al., 2006), but the exponent takes universal values across different basins (i.e., exponential parameter $b = 0.3$, proportional parameter $a = 0.01$) for convenient global applications. It takes the 250-m SRTM DTM as terrain input and implements the hydrological analysis workflow by using the ArcPy library.

*(4) GLWD lake and reservoir dataset.* We apply a global lake mask to crop the reference map before using it for parameter estimation, which helps to avoid inconsistent lake representations from our reference and validation datasets. To do that, the Global Lake and Wetland Dataset (GLWD) jointly developed by the World Wildlife Fund (WWF) and the Center for Environmental Systems Research at the University of Kassel (Lehner and Döll, 2004) is used. It consists of three layers and the Level-1 layer represents large lakes and reservoirs, including 3,067 lakes and 654 reservoirs with lake area $\geq 50$ km² and storage capacity $\geq 0.5$ km³, respectively. The dataset takes reference from multiple sources and is further refined with independent data from USGS and extensive visual inspections and quality control.

### 2.1.3 Datasets for correlation

*(1) Global-AI_PET_v3 Aridity Index database.* We use the Aridity Index (AI) from the Global Aridity Index and Potential Evapotranspiration Database (Global-AI_PET_v3) to assess its linkage with the FHG parameter. The database provides 30 arc-second global Potential Evapotranspiration ($ET_0$) and AI data. AI is calculated as the ratio of mean annual precipitation to mean annual reference $ET_0$, which is estimated by the FAO Penman-Monteith Reference Evapotranspiration equation. It has been validated against various weather station data and shows an improved correlation with real-world data compared to previous versions (Zomer et al., 2022).

*(2) LAI Climatology.* Developed for a model intercomparison project (HighResMIP v1.0) of CMIP6, this dataset provides a global 0.25° x 0.25° gridded monthly mean leaf area index (LAI) climatology, averaged from August 1981 to August 2015 (Haarsma et al., 2016). Derived from the Advanced Very High Resolution Radiometer (AVHRR) Global Inventory Modeling and Mapping Studies (GIMMS) LAI3g version 2, it includes bi-weekly data from 1981 to 2015. The raw LAI3g version 2 data were regridded from 1/12° x 1/12° to 0.25° x 0.25°, processed to remove missing and unreasonable values, scaled to obtain LAI values, and averaged bi-weekly to monthly. The final product is a monthly long-term mean LAI (1981-2015) provided in a single NetCDF (.nc4) file.

## 2.2 Methods

This section introduces HAND, FHG and our parameter estimation scheme for SHIFT.

*(1) HAND as a terrain attribute.* HAND is a derivative terrain index that describes the relative elevation difference between any grid cell in a DEM and its nearest river grid (Rennó et al., 2008). River grid here is identified by applying a 1000 km$^2$ threshold to the Upstream Drainage Area (UPA), supported by previous studies (Nardi et al., 2019). The threshold is determined by preliminary experiments to ensure that it is neither too small, which would misattribute large-river-dominated floodplains to small rivers, nor too large, which would overlook rivers with notable influence. Accurate HAND calculation requires defining the nearest river network grids either by flow direction or by distance. The flow direction model defines the first river network grid reached by tracing the d-8 flow as the nearest drainage, resulting in floodplain maps that capture regional hydrological characteristics but influenced by local terrain fluctuations. The distance model searches for the nearest drainage grid with a specific distance (e.g., two-dimensional or three-dimensional Euclidean distance), highlighting geometric considerations but ignores natural geomorphic separations. We adopt the flow direction method to avoid discontinuities in HAND introduced by the distance model (not shown); subsequent results in floodplain delineation are derived from using the d-8 flow directions obtained from the MERIT-Hydro dataset.

*(2) FHG as a thresholding scheme.* FHG is an adapted form of the original river channel hydraulic geometry (Leopold and Maddock, 1953). It posits a power-law relationship between floodplain characteristics (width, depth, 100-year discharge) and river size (UPA or Strahler stream order). In the context of floodplain delineation, it considers a power-law relationship between the potential inundation depth (*h*) of a river grid cell and its UPA:

$$h = a \cdot UPA^b \hspace{4cm} Eqn.\,(1)$$

where *a* and *b* are empirical parameters containing heterogeneous factors determining floodplain extents. Then, the algorithm determines grid cells with HAND lower than *h* of the corresponding river grid ($h_{river}$) as floodplain, which can be represented by Eqn. (2):

$$f(HAND, h_{river}) = \begin{cases} 1, HAND \leq h_{river} \\ 0, HAND > h_{river} \end{cases} \hspace{2cm} Eqn.\,(2)$$

*(3) FHG parameter estimation.* Estimating FHG parameters requires either reference floodplain extents or estimated runoff as inputs (Annis et al., 2019; Nardi et al., 2013). We take two hydrodynamic model outputs as the reference map (i.e., the 500-year return period JRC and GAR flood maps) as they intrinsically contain floodplain spatial heterogeneity by feeding gauged streamflow observations or climate reanalysis data (Lindersson et al., 2021). The goal of using these reference datasets is to capture the information of spatial heterogeneity of these two datasets while trying our best to constrain model-related uncertainties.

With the above reference map, two methods can be used to estimate FHG parameters: parameter space sampling (PSS) and logarithmic regression (LR). PSS defines a feasible range for two parameters in Eqn. (1), and then samples the parameters from the parameter space and tests their combinations against a reference map by using a fitness index (Annis et al., 2019). LR assumes that all floodplain grids from the reference map satisfy the scaling law so that the FHG parameters *b* and *a* can be estimated by statistical approximation (Nardi et al., 2013). For LR, we expect all floodplain pixels in the reference map to satisfy:

$$HAND \leq a \cdot UPA_{river}^b \hspace{3.5cm} Eqn.\,(3)$$

which could be transformed into:


$$ln(HAND) \leq b \cdot ln\,(UPA_{river}) + ln\,(a) \qquad\qquad Eqn.\,(4)$$

Apparently, PSS can best approximate the output but could lead to equifinality, while LR emphasizes the scaling law but could be influenced by uncertainties of the reference data (see more

details below). Therefore, we combine the two methods above and propose a stepwise parameter estimation framework. Specifically, we first determine baseline values for parameter $a_0$ from prior research, then estimate $b$ by forcing logarithmic regression based on the reference dataset to best respect the FHG scaling law; then, the coefficient $a$ is calculated by sampling parameter space based on the reference map and the determined $b$ value.


Equation (4) anticipates a positive linear relationship (**Fig. 2a**) between $ln(UPA_{river})$ and the maximum $ln(HAND)$ values that mark the furthest floodable grid by each river grid, but our observations deviate from this expectation because some river grids with small drainage areas can have unexpectedly high HAND values. These can be ascribed to uncertainties with our reference

map that inherits the model chain errors, terrain data and spatial resolution inconsistencies, as well as other unaccounted within-basin variability that may break the scaling law. While results from the Gaussian kernel density plot (**Fig. 2b**) prove that the majority of data still conforms to the power law, the patterns indicates that one cannot simply apply LR to the maximum $ln(HAND)$ and $ln(UPA_{river})$ to obtain the required parameters.

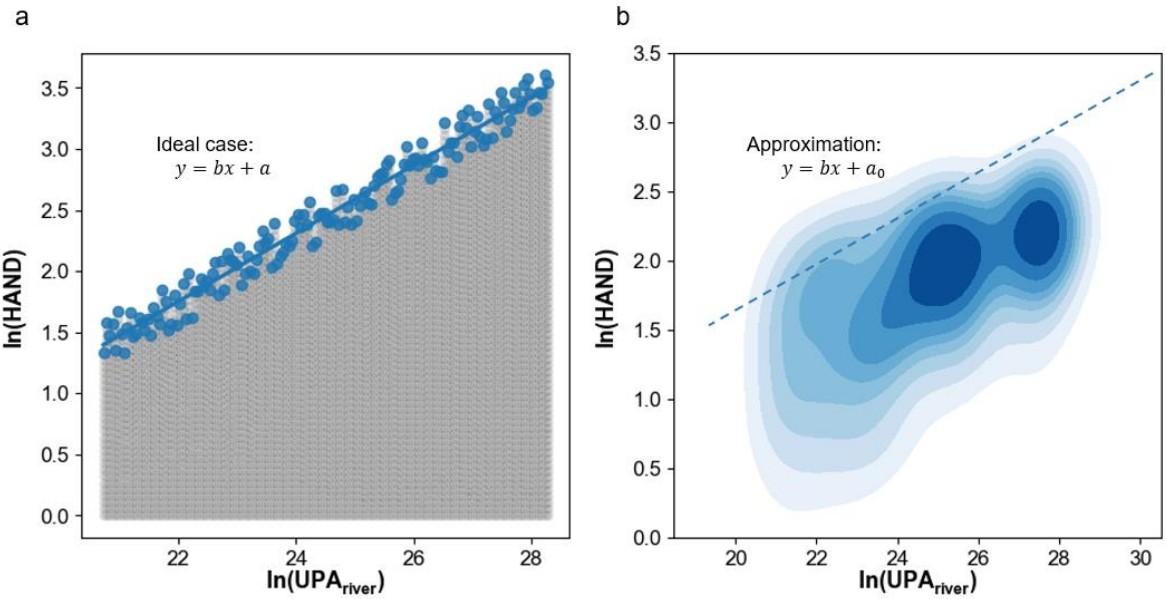


**Figure 2. The expected and actual scenario of floodplain grids within a basin**. X-axis represents $\ln(UPA_{river})$, y-axis represents $\ln(HAND)$. a) shows the scatter plot of the expected linear relationship between maximum $\ln(HAND)$ and $\ln(UPA_{river})$. Blue scatters are the maximum HAND values while gray scatters are non-maximum reference floodplain grids. b) shows an actual

scenario (i.e., the Yangtze River basin corresponding to the level-3 HydroBASINS, PFAF ID: 434) which approximately arrives at the power law relationship in kernel density plot.

Therefore, we develop a scheme to effectively mitigate data noises in estimating parameter *b* while maintaining the power law for the majority of grids. First, we take floodplain grids from the

intersection of the two reference maps as we suggest the intersection map to be more accurate. Then we set a universal HAND threshold of 20 m to screen out the most obvious high anomalies. We then group HAND values by $UPA_{river}$ and apply an iterative moving-window data-filtering scheme based on 3-sigma statistics where every grid would be filtered by 20 windows (window size = 1, step = 0.1). In each iteration, we compute the mean and standard deviation for the data

within each window. A grid point is retained only if it consistently meets the 3-sigma criteria across all 20 windows. This iterative process stops either when every data point fits within all moving windows, or if the procedure fails to converge towards a stable solution (e.g., for highly noised or significantly non-normal data). Instead of directly performing LR, we calculate a sequence of theoretical *b* values from the maximum HAND of each $UPA_{river}$ unit with a baseline estimate of $a_0$

= 0.01 based on prior research (Nardi et al., 2019).The binning parameter is tuned to effectively reduce data noise for all basins. As the optimal *b* will lean towards the higher end of our calculated sequence but not at the highest end as it could be possibly interfered by remaining high HAND anomalies, we evaluate the 10% to 50% percentiles of these *b* sequences across all basins to identify the best percentile that centers around the previously estimated global *b* value of 0.3 (Nardi

et al., 2019). The *b* value under this identified percentile is then chosen as the optimal parameter for each individual basin.

After *b* is determined, the coefficient *a* is optimized with an iterative PSS method. We take both hydrodynamic maps as our reference dataset, as we would like to highlight the 'consensus' of

existing maps while trying to achieve better consistency from both maps. Numerous indices for the optimization target exist, including Overall Accuracy (OA), Kappa coefficient (Cohen, 1960), Fleiss's Kappa (Fleiss, 1971), Model Agreement Index (MAI, Trigg et al., 2016) or measure-of-fit function (Nardi et al., 2019). While MAI and measure-of-fit function emphasize data overlap, they do not address overprediction. OA considers unpredicted areas but may overly reward non-

floodplains since they are the major landmass type. Fleiss's Kappa can assess agreement among multiple datasets, but using it alone with two existing datasets may bias our estimated boundary towards the dataset with larger predictions as it maximizes mathematical consistency values. This is undesirable since we aim for agreement with each individual dataset to balance the information from both. Therefore, our target function is defined as:


$$Consistency \ = \ FK - \sigma \cdot \left( MAI_{JRC} - MAI_{GAR} \right)^2 \qquad\qquad Eqn.\,(5)$$

Here, Fleiss's Kappa (FK) represents how well the three datasets (including the product with the parameter to be optimized) match. The penalty term, based on the squared difference between the

two MAI values, reduces bias towards one dataset over the other. The weight term (σ), ranging from 0 to 1, is determined by the normalized number of available reference data grids in the basin. This assumes that basins with fewer common data grids have less reliable datasets, and thus overemphasizing the penalty term would unnecessarily and overly influence FK.

Fleiss's Kappa (*FK*) is calculated as:

$$FK = \frac{P_o - P_e}{1 - P_e} \qquad \qquad Eqn.\,(6)$$

Where $P_o$ and $P_e$ are respectively calculated by:


$$P_o = \frac{1}{N}\sum_{i=1}^{N}\frac{\sum_{j=1}^{C} n_{ij}(n_{ij} - 1)}{K(K - 1)} \qquad \qquad Eqn.\,(7)$$

$$P_e = \sum_{j=1}^{C}\left(\frac{\sum_{i=1}^{K} n_{ij}}{NK}\right)^2 \qquad \qquad Eqn.\,(8)$$

In these equations, $K$ is the number of models (3 here), $N$ is the number of pixels, $i$ represents each

grid, and $j$ represents different possible values (1 or 0 here). The MAI is calculated as:

$$MAI = \frac{A}{A + B + C} \qquad \qquad Eqn.\,(9)$$

Where $A$, $B$, $C$ denote overlapping (True Positive), over-prediction (False Positive), and under-

prediction (False Negative) respectively. Considering the previously estimated $a$ values range
across 0.001 to 0.06 (Nardi et al., 2018), we first sample 20 equidistant $a$ values between 0 and 1
against the reference data. Then the direct neighbor of the best-performing $a$ value, constraining
its precision to at least one decimal place, is used to search for the true optimal $a$. We apply five
iterations, each with a new set of 20 equidistant $a$ values within the estimated direct neighbors

from the last iteration. The optimal $a$ from the final iteration is then selected as the basin-specific
coefficient.

*(4) Development of SHIFT.* Based on the above, we estimate the FHG parameter with the
HydroBASINS level-3 basins, and derive the floodplain map for each basin and then integrate

them into a 90-m global floodplain map. We use Python 3.10 libraries (e.g., Pandas, Numpy and
Geopandas), the GDAL command line interface and the TauDEM toolkit (Tarboton, 2016) for the
FHG parameter estimation and the thresholding . We also downsize the dataset to 1-km resolution

floodplain map for convenient large-scale applications - the 1-km resolution floodplain map is provided as part of the final output. We used median as the resampling method for continuous variables like UPA and HAND, and mode for categorical data, such as the reference maps, SHIFT, and watershed division. Permanent water bodies are removed for all processes.

*(5) Validation & Correlation.* After getting the updated floodplain boundary with the optimized parameters (SHIFT), we conduct a pairwise consistency analysis among five maps, i.e., SHIFT, GFPlain250m, UP (Universal Parameters, applying $b = 0.3$ and $a = 0.01$ on MERIT-Hydro), JRC and GAR. UP was generated to allow the assessment of how changes in parameters influence the results. We apply both MAI (see Eqn. 9) and OA for this pairwise consistency analysis in reference to previous research (Lindersson et al., 2021). Note that MAI is a critical index: a MAI of 0.2 represents 20% to 33% overlapping between models, while a MAI of 0.5 represents 50% to 67% overlapping. Previous large-scale assessment of floodplain map consistencies revealed that the median MAI is at the range of 0.1 to 0.4 (Lindersson et al. 2021). OA is calculated as:

$$OA = \frac{A + D}{A + B + C + D} \qquad\qquad Eqn.\,(10)$$

where $D$ denotes non-prediction by both maps, and $A$, $B$, and $C$ are as defined in Eqn. (9). The two types of indices applied here have different focuses: OA considers non-floodplain areas, while MAI focuses exclusively on overlapping floodplain areas. Considering the overall landmass is non-floodplain, we also calculated OA within 20-km buffer zones, with distance measured as the hydrological distance to the stream. In the pairwise comparison, group comparisons were conducted with JRC and GAR, where each hydrodynamic map was tested against SHIFT, GFPlain, and UP. The JRC-GAR pair serves as the baseline.

For our analysis, we focus on parameter $b$. Theoretically, $b$ influences rivers differently based on their drainage area, with larger $b$ values highlighting the dominance of large rivers over tributaries in shaping floodplain extents. Thus, we expect $b$ to be closely associated with the spatial heterogeneity of basin-level hydrological and geomorphic characteristics. Our primary hypothesis is that $b$ should be related to climate aridity, as more humid areas are expected to show stronger

dominance of large rivers. Additionally, vegetation, indicated by LAI (Leaf Area Index), may also play a role since it is involved in the runoff generation process as well as modulating soil erosion

that can be key to floodplain formation. Therefore, we calculate the correlation of $b$ with average AI (Aridity Index) and LAI across all basins to validate whether our thresholding scheme can better capture the spatial heterogeneity of floodplain characteristics.

## 3. Results


### 3.1 Global FHG parameter estimation

Following the stepwise parameter estimation scheme proposed in Section 2.2 (3), we obtain the statistical distributions of different $b$ percentiles in **Fig. 3b**. While distributions from all percentiles exhibit similar patterns, especially for the 20th to 50th percentiles, we apply the 30th percentile

worldwide as it best distributes around the previously estimated global $b$ value of 0.3 (Nardi et al., 2019; see dashed line in **Fig. 3b**). The majority of estimated parameter $b$ lies within the range of 0.25 and 0.35. Based on the estimation, the coefficient $a$ is also optimized which varies from 0.0001 to a maximum of 0.12 across all basins.

Spatially, the distribution of the estimated $b$ (**Fig. 3a**) shows that regions characterized by abundant precipitation and water resources (e.g., southern East Asia, Southeast Asia, the Mississippi and Amazon) generally exhibit relatively higher $b$ values. Conversely, regions such as Central-West Asia, the Arabian Peninsula, the Sahara region, and central Australia tend to have relatively lower $b$ values. There are also exceptions, for instance the overall high $b$ values in the arctic circle, and

low values in river deltas (e.g., the western Mississippi Delta in Louisiana, Jiaodong Peninsula in Eastern Asia). Estimated parameter $a$ exhibits a less clear spatial pattern (**Fig. S1**) as it is less uniform in unit and highly dependent on estimated $b$.

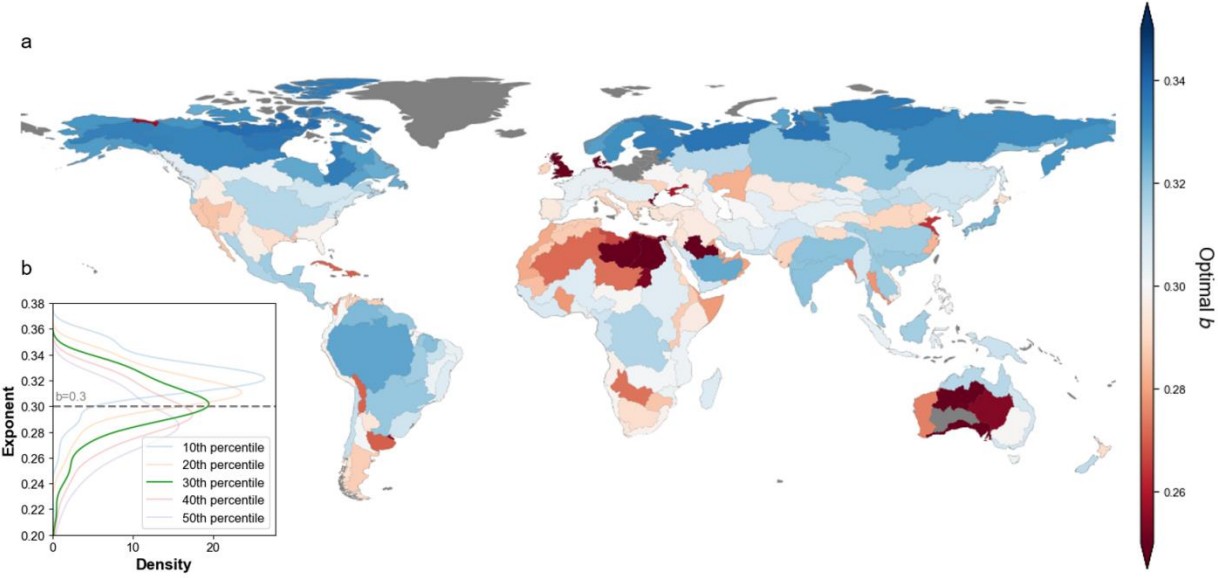

**Figure 3. Statistical and spatial distribution of estimated FHG parameter b.** a) Spatial distribution of parameter *b* across HydroBASINS level-3 basins. b) The distribution of parameter *b* across basins, $p_{10}$ to $p_{50}$ represent the percentiles during estimation, *b* = 0.3 line shows the universal value applied in previous research.

Statistically, results show that *b* from all basins exhibits significant but weak positive correlation with the AI (Aridity Index, *r* = 0.335, **Fig. 4a)**, and an insignificant positive correlation with LAI (Leaf Area Index, *r* = 0.083, **Fig. 4b**). For the selected 33 major basins, which are hydrologically connected and thus expected to have more internally consistent hydrological characteristics, both correlations are stronger (*r* = 0.680 for AI and *r* = 0.668 for LAI) and significant. We investigate other potentially relevant factors (**Fig. S1**), but no significant and consistent linear correlations were observed, suggesting that more complex mechanisms may be involved that do not manifest as observable linear correlations.

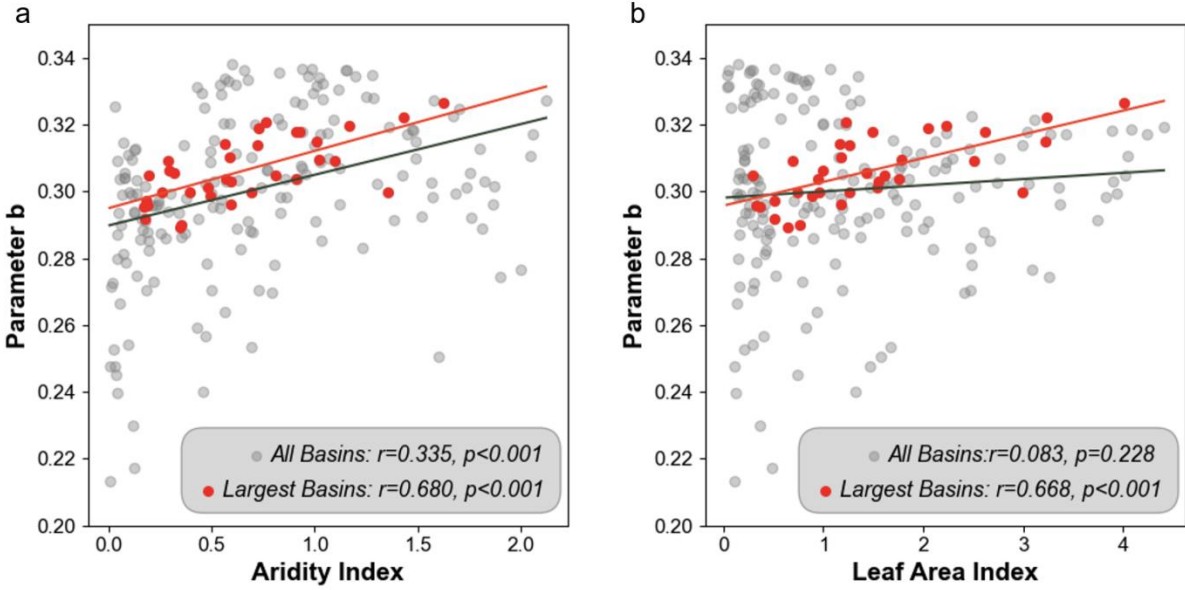

**Figure 4. Scatter plots of FHG parameter b against relevant hydroclimatic factors.** a) Scatter plot with AI (Aridity Index). b) Scatter plot with LAI (Leaf Area Index). In each plot, gray points represent all basins, including the largest ones, while red points represent the 33 selected basins. Pearson's *r* and significance levels are indicated on the plots.

## 3.2 Global floodplain delineation

Based on the estimated FHG parameters, the global distribution of floodplain areas is delineated and shown in **Fig. 5a**. Overall, the spatial pattern of the floodplains aligns well with the low-lying areas in major river systems. More specifically, floodplains in Northern Asia are mainly distributed around the West Siberian Plain and the Central-West Siberian Plateau, e.g., the Ob and Yenisei River Basins. West and Central Asia's floodplains are primarily near the Caspian Sea, the Aral Sea, and the Mesopotamian Plain. In East Asia, the Yangtze River Basin dominates floodplains in the middle and lower reaches, along with contributions from the North China Plain, some Yellow River tributaries such as the Hetao Plain, and river mouths in the southeast. The Lancang-Mekong River Basin and the Salween-Irrawaddy River Basin in Southeast Asia also breed the largest floodplains worldwide, as well as the Indus and Ganges-Brahmaputra River Basins from South Asia. In Europe, the primary floodplains are concentrated in the Danube River Basin between the Alps and the Carpathian Mountains, alongside the Rhine, Dnieper and Po River Basins. In Africa, floodplains

are predominantly distributed in the upstream Nile River, including the Nile Delta and the Niger

River Basin, as well as the Congo River Basin, the Chari River-Lake Chad Basin and around Lake

Victoria, with additional areas near West and East Africa's coasts. North America's floodplains are

mainly in the Mississippi River Basin and Alaska's Yukon River Basin. South America's

floodplains are primarily in the Amazon River Basin, the Orinoco plain, and the La Plata plain. In

Oceania, floodplains center in the interior lowlands around the Murray-Darling River Basin.


To show more regional details, we use two cases under different climatic conditions (**Figs. 5b &**

**5c**) to further illustrate the differences between SHIFT and the widely used GFPlain250m dataset.

Case 1 (**Fig. 5b**) is the Indus-Ganges-Brahmaputra River basin which flows through Bangladesh,

India, Pakistan, and Nepal. These countries are primarily characterized by frequent floods and are

strongly influenced by the South Asia monsoon. SHIFT captures detailed floodplains in the Indus

River basin, a major basin in South Asia where GFPlain250m leaves out. Additionally, SHIFT

offers finer details in upstream areas and can better distinguish main river floodplains from those

of the tributaries. Case 2 (**Fig. 5c**) is situated in the Yellow River basin (Hetao Plain) in Inner

Mongolia, China, a region dominated by arid to semi-arid continental climate. Comparing the

floodplain maps with visual interpretations of the satellite images suggests that SHIFT can provide

a more comprehensive depiction of the Hetao Plain. The floodplains outside of the Hetao Plain in

SHIFT are relatively limited, which aligns with its generally dry climate conditions.

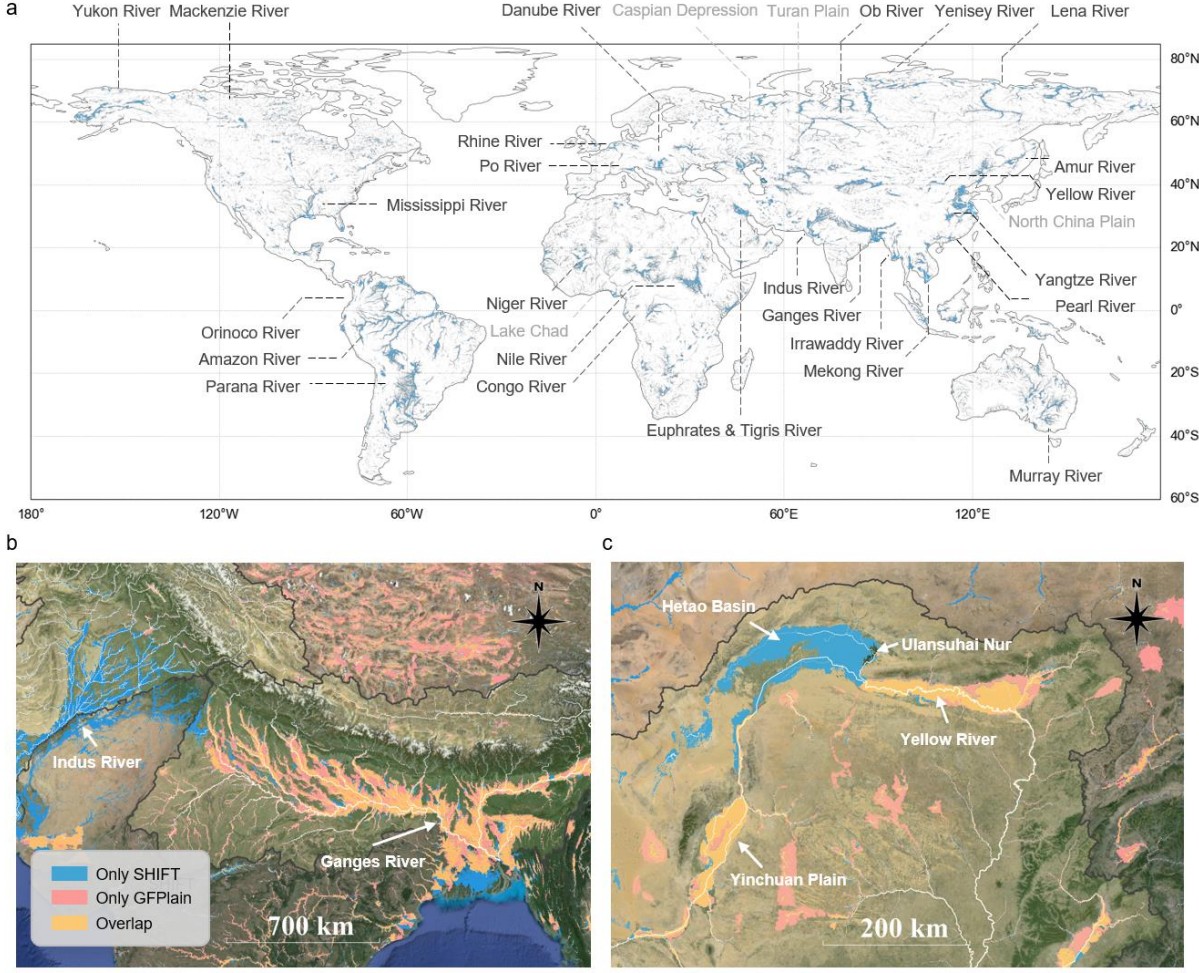

**Figure 5. Geomorphic floodplain extent in SHIFT**. a) Global spatial distribution of floodplains, with major river basins or plains marked out. b) and c) show two cases of that compares SHIFT with GFPlain250m, with background image from © Google Earth on EPSG: 3857 projection. b) locates in the humid Indian-Ganges River basin, while c) locates in the semi-arid yellow river basin in inner Mongolia, China. Major rivers of the region is marked on the map. SHIFT delineates fewer areas in the upstream Ganges River (b) and reduces the floodplain extent outside the Yellow River mainstream (c). It also offers more comprehensive coverage, including the Indus River basin (b) and the Hetao basin (c).

According to SHIFT, global floodplains take up approximately 9.91 million km², representing 6.6% of the world's total land area. **Fig. 6** further shows the floodplain area and the percentage of

floodplains within each of the global major river basins. Overall, the Amazon River Basin possesses the largest total floodplain area globally (625,431.3 km$^2$), followed by the Parana, Nile, Ganges and Mississippi River basins. Floodplains in Haihe River Basin takes up the greatest area

percentage (~20%), highlighting the great geomorphic flood inundation potential of such basins. Comparing across continents, South America and Asia breed the most widespread floodplain extent worldwide and tend to have the highest floodplain percentages.

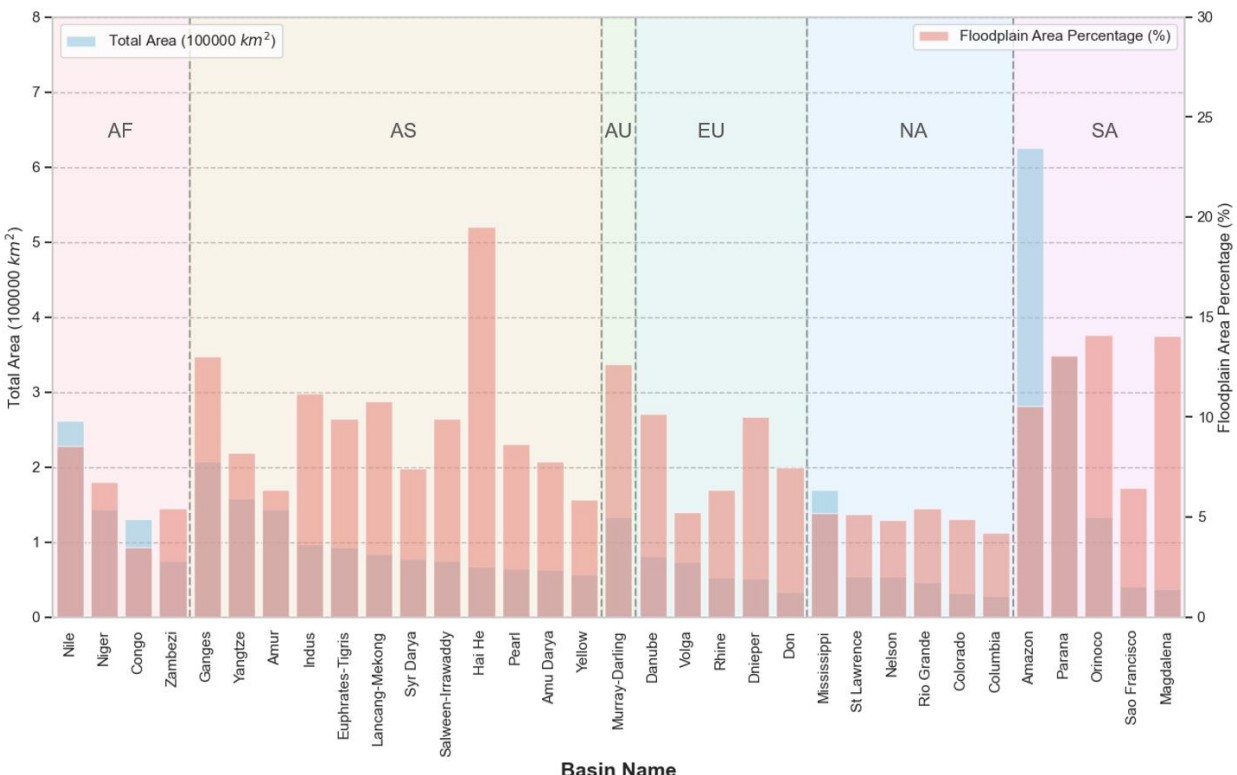

**Figure 6. Floodplain area statistics in major river basins**. Blue bars stand for total floodplain area (left y-axis), red bars stand for the ratio of floodplain area to the total basin area (right y-axis). Basins are ranked by total floodplain area. AF: Africa, AS: Asia, AU: Australia, EU: Europe, NA: North America, SA: South America.


### 3.3 Validation and consistency analysis

**Figure 7** shows the basin-level distribution of MAI between SHIFT and the two hydrodynamic maps. It shows that: (1) SHIFT exhibits stabler consistency with the two maps in major basins

(e.g., the Yangtze and the Amazon) compared to smaller basins; (2) Better consistency between
SHIFT and reference maps is found in humid basins, with some exceptions in arid areas (e.g., the
Niger). These patterns may be attributed to the greater number of reference data grids in larger and
wetter basins, which also have strong scaling relationships that support a geomorphic approach for
floodplain mapping. In addition, SHIFT generally aligns better with JRC in major river basins,
while consistency with GAR is higher in smaller basins and inland river basins. The median MAI
with JRC is 0.271, while for GAR it is 0.308. For the 33 major basins, the median MAI with JRC
and GAR are 0.415 and 0.289 respectively. This difference can be ascribed to the different river
stream delineation strategies adopted by the two datasets. That is, JRC uses a stream threshold of
5000 km² for drainage area, while GAR uses 1000 km². Consequently, JRC is less effective in
capturing features of inland basins (e.g., the Tibetan Plateau) and fragmented river deltas (e.g.,
west of the Andes), where few rivers meet the 5000 km² threshold. For large basins, JRC performs
better as it highlights the inundation of larger rivers (e.g., the Mississippi), while in GAR small
rivers also yield large floodplain extents. Notably, SHIFT generally aligns better with JRC for the
Arctic basins, as GAR lacks data north of 60°N.

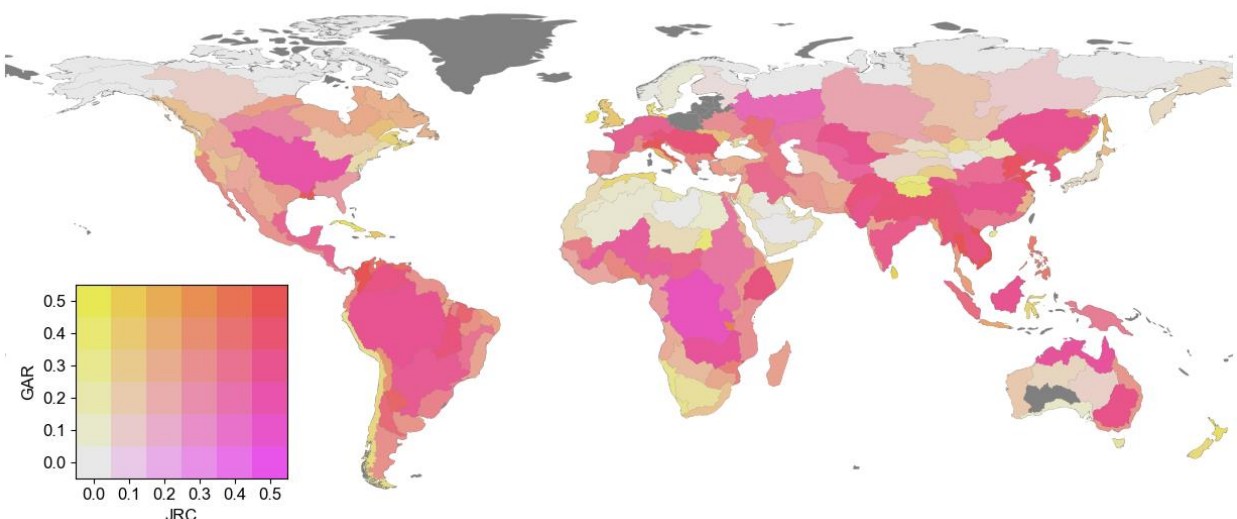


**Figure 7. Validation of SHIFT against two reference datasets.** In the bivariate map, the two
variables are the MAI against the JRC map (magenta) and the GAR map (yellow). A balanced
MAI results in red basins.

**Figure 8a** shows the pairwise consistency analysis among different floodplain maps to more objectively document the pros and cons of each dataset. Prominently, it shows that the consistency between SHIFT and JRC significantly improves over UP and GFPlain, but that with GAR does not (as shown in MAI). The consistency pattern can be explained by delving into the inner working of each dataset. For large basins, SHIFT highlights the mainstreams and reduces prediction of

tributaries, thus aligning more closely with JRC as it highlights major rivers, leading to a decrease in consistency with GAR. UP and GFPlain align better with GAR in these regions, as they all tend to overpredict, especially in tributaries. For other basins, SHIFT strikes a balance between the two datasets. Comparing SHIFT with UP, SHIFT increases the lower interquartile range for JRC's OA and the upper interquartile range for GAR's OA, highlighting a general improvement with SHIFT.

For MAI, the upper quartile with GAR has decreased while the lower quartile has improved, suggesting a consistency trade-off between the two datasets. Notably, all geomorphic maps show a better consistency with the hydrodynamic outputs than the hydrodynamic pair, proving again that the hydrogeomorphic delineation method is a more globally consistent framework.

To better understand the impact of our estimated parameters on the consistency performance, we analyze the most consistent pair and corresponding MAI values for each basin. Among all pairs, SHIFT-JRC aligns the best in 62 basins, with SHIFT-GAR in 74, UP-JRC in 8, and UP-GAR in 37 (**Fig. 8b**). This validates that SHIFT exhibits better consistencies with the reference maps even though the difference between SHIFT and UP seems not statistically significant (**Fig. 8a**). Spatial

patterns (**Fig. 8b**) show that SHIFT-JRC pairs aligns best in humid major basins (e.g., the Mississippi and Amazon) and very arid regions (e.g., the Taklamakan and central Australia). SHIFT-GAR pairs are the most consistent in mountainous regions (e.g., the Rockies and Andes), aggregated deltas (e.g., eastern Australia and southern Africa), islands (e.g., Indonesia), and inland river basins (e.g., the Tibetan Plateau) where few rivers meet the 5000 km² drainage area threshold

of JRC. In contrast, cases where UP pairs align best are less common. UP aligns better with GAR due to their shared large prediction extents, such as around the Caspian Sea. In rare instances where UP-JRC pairs perform best, it is typically in deltas or regions where SHIFT-GAR performs well, such as deltas and islands. This is likely because our method balances consistency between the datasets, but GAR's wider prediction coverage makes this strategy less effective in these infrequent

cases.

Note that GFPlain and UP use the same parameter for its geomorphic delineation, but their consistency with JRC and GAR differs significantly (**Fig. 8a**). This is because GFPlain uses 250-m SRTM as the terrain input, while UP uses MERIT-Hydro, which has undergone hydrological

correction to lower the elevation of waterbody pixels, resulting in higher HAND values and smaller floodplain extents. GAR, which generally overpredicts floodplain extents especially in arid regions, aligns better with GFPlain. The overprediction of GAR is evidenced by GAR-pairs having the lowest OA, as OA strictly penalizes overprediction. At the same time, we found the difference between SHIFT and UP may be under-represented in the statistical plots (**Fig. 8a**) while the actual

impact of variable parameters brought by SHIFT is substantial: the global floodplain extent estimates are 14.95 million km² for UP and 9.91 million km² for SHIFT, showing a 50.85% difference in total predicted areas. Additionally, regions where UP-GAR has the highest consistency (**Fig. 8b**) generally coincide with regions where SHIFT-JRC aligns best. This reversed pattern of consistency further supports that the statistical differences between UP and SHIFT are

underrepresented in **Fig. 8a**.

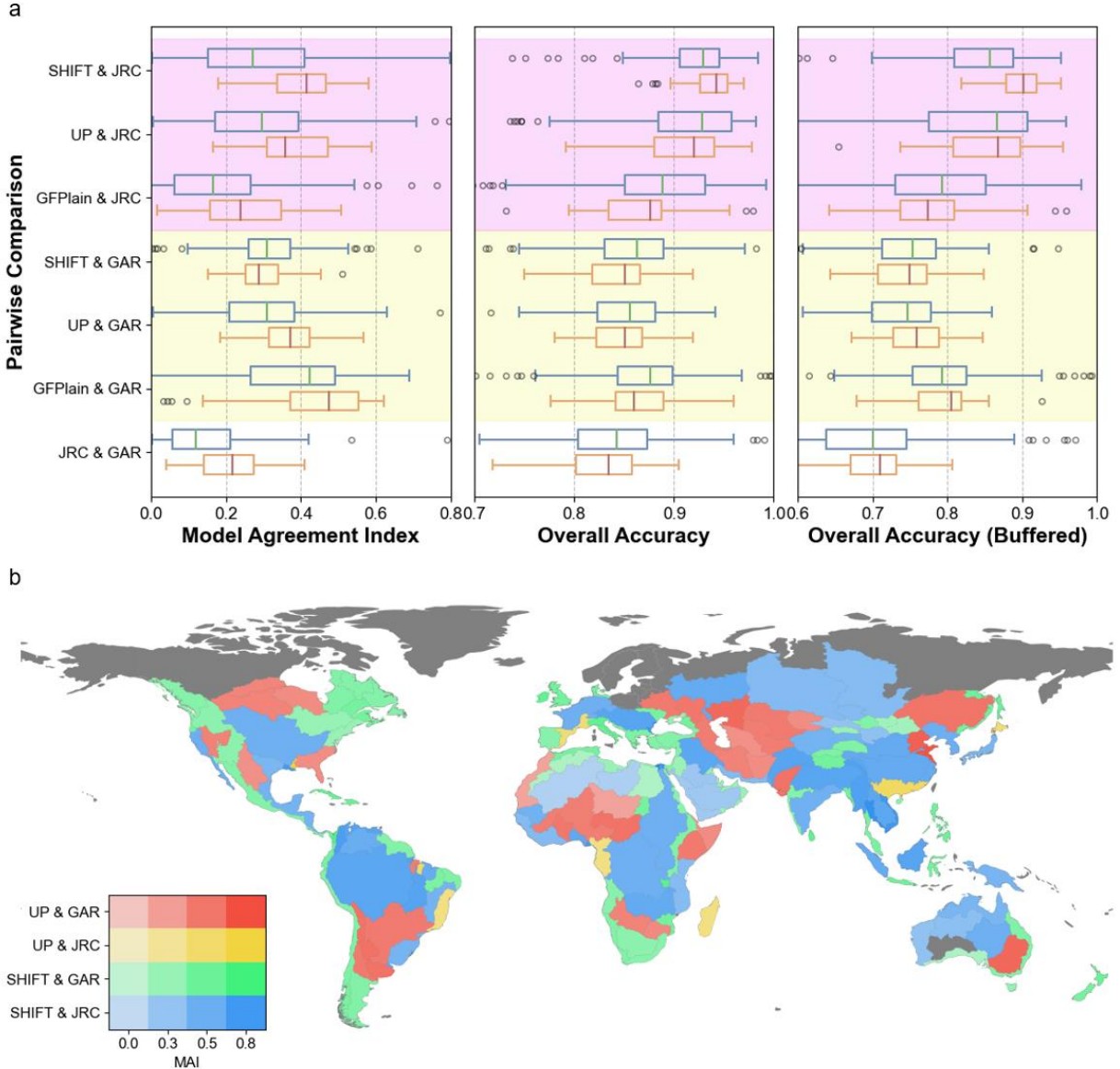

**Figure 8. Results of the consistency analysis.** a) Boxplots of pairwise analysis among SHIFT, GFPlain, UP (MERIT-Hydro but with Universal Parameters), JRC, and GAR across three metrics: MAI (left), OA (middle), and OA within a 20-km buffer (right). Two group comparisons are marked in different colors (magenta for JRC and yellow for GAR). Statistics for all basins with valid data inputs (see Methods) are shown in blue boxes, and those for the 33 major river basins are shown in orange. b) Bivariate choropleth map of the highest-performance MAI pair among four pairs (SHIFT & JRC, SHIFT & GAR, UP & JRC, UP & GAR) and the corresponding MAI value for each basin. Different pairs are represented by different hues, with higher MAI values

shown in higher saturation. Basins where a SHIFT-pair performs best are marked in cold colors, while those where a UP-pair performs best are represented in warm colors. Among all pairs, SHIFT-JRC performed best in 62 basins, SHIFT-GAR in 74, UP-JRC in 8, and UP-GAR in 37.

# 4. Discussions

Several conceptual and technical details warrant discussions when developing our improved geomorphic parameter estimation approach. Here, we discuss the pros and cons of SHIFT with respect to its FHG thresholding scheme, residual uncertainty, hydrogeomorphic floodplain boundary, and the spatial scale used in our methodological development.

## 4.1 FHG as a thresholding scheme

The primary contribution of this study is the estimation of localized parameters for the FHG model. In section 3.3, we compared the performance of localized versus global parameters, but several aspects require further clarification.

First, we believe the need for localized parameters arises from the role that empirical parameter in FHG plays in determining floodplain boundary. A higher $b$ value emphasizes the influence of larger rivers in shaping geomorphic floodplains, reflecting hydrogeomorphic processes that vary across different basins and should be better represented. Given the absence of ground truth for floodplain boundaries, we attempt to improve representation of these heterogeneous processes by balancing information from two existing reference maps from hydrodynamic modeling. Despite acknowledged inconsistencies, the hydrodynamic maps are informed by climatic forcing, providing a common basis more likely to spatially heterogeneous than universal geomorphic parameters. In other words, while we do acknowledge these maps can be uncertain, they contain useful information that can be applied to constrain geomorphic floodplain boundaries. This leads to our data filtering process to reduce inconsistency and to identify a scaling law from the references. By incorporating outputs from hydrodynamic maps, our approach optimizes the DEM-based model without altering its foundation, as evidenced by the overall better consistency regardless of parameters used (**Fig. 8a**). Although certain regions may benefit less from our

strategy (e.g., where UP-JRC performs best), results (**Fig. 8b**) show convincing general improvements and consistency patterns. The estimated parameters derived here are also provided

to support potential future studies with regionalized focuses.

Second, our estimated parameters aim to capture fingerprints from spatially varying hydrological and geomorphic processes that can influence the floodplain extent. We consider aridity as the primary factor influencing the spatial variability of $b$, based on the assumption that in humid basins,

rivers with larger upstream drainage areas exert greater dominance over smaller segments in shaping floodplains. Vegetation also plays a role, as it influences runoff generation and modulates soil erosion, both key to floodplain formation. Additionally, factors such as terrain and soil composition might influence the results. Given the data uncertainties and the complex physical interpretations of $b$, it is important to note that we do not expect perfect relationships between these

factors and the derived exponent $b$. The correlation analysis indeed aligns with our expectations: AI is statistically significant in explaining the spatial variability of $b$, while LAI plays a role and terrain does not show strong correlations with $b$. Soil compositions (Poggio et al., 2021) do not exhibit a consistent pattern across analyses done at different scales (**Table S1**). Despite the not-so-strong correlation with AI and LAI, its statistical significance supports the effectiveness of our

proposed methods, which helps to derive spatially-varying parameters that are also physically meaningful. The parameter $a$ could also encapsulate influences from relevant processes, but its physical interpretation is highly dependent on $b$, as its unit is less uniform (Nardi et al., 2006). Therefore, clarifying the influencing processes of $a$ is beyond the scope of this study.

Third, although alternative thresholding methods that use river discharge and synthetic rating curves exist (e.g., those used by the US National Water Model, Zheng et al., 2018), these methods come with more sources of uncertainty by requiring high-quality data inputs (e.g., gauged discharge, Manning's coefficient). Thus, while they may work well with in-situ observations, replicating this globally poses challenges and is conceptually different from our approach. Our

proposed FHG method requires only terrain input, which is recognized as the least uncertain component in global floodplain mapping method (Bates, 2023). By providing the optimized parameters derived here, we consider the FHG thresholding as more globally consistent and easily applicable.

**4.2 Residual uncertainties associated with FHG parameter estimation**

We also recognize several uncertainties associated with the FHG relation. The primary source of uncertainty comes from the inconsistency between the two reference hydrodynamic datasets across regions, which can be traced back to their model chain errors. Several measures are taken to mitigate the potential influence: we take the intersection of the two datasets as the reference, apply

an iterative moving-window scheme to filter the data, and force scaling-law relationships to estimate the parameter $b$. However, residual uncertainties may still exist due to three aspects: (1) Inconsistencies in terrain data, as both JRC and GAR use SRTM as the inputs while we use MERIT-Hydro; (2) Potential intra-basin heterogeneity of scaling relationships which may lead to unstable estimates; (3) The lack of reference data in certain basins, which lowers our credibility in

the estimated parameters. To evaluate how the residual uncertainty influences our FHG parameter estimation, we quantify the uncertainty of $b$ by calculating the standard deviation among all possible $b$ values derived at different percentiles. This metric assesses how well the data conforms to the power law: a better-conforming set of data result in a narrower range of the estimated $b$ sequence and, consequently, lower standard deviation. A lower standard deviation also supports

the application uniform filtering percentiles globally (see Methods) and proves robustness of our approach.

**Figure 9** reveals the residual uncertainty in parameter $b$, which ranges from 0 to 0.03 with a median of 0.01. This is considered reasonable for a global median $b$ of 0.3. The pattern is similar to that

parameter $b$ itself (**Fig. 3**), with lower uncertainties in large humid basins (blue color), and greatest uncertainty (red color) observed in arid regions (e.g., The Saharan Regions and western-central Australia), mountainous areas (e.g., the Rocky Mountains and the Andes), and deltas (e.g., The Jiaodong Peninsula, the western Mississippi Delta and the Nile Delta). High residual uncertainty in these regions are possibly due to the particularly strong differences between the reference

datasets. For deltas, the great inconsistencies in spatial extents are amplified by their different definition of rivers, as JRC and GAR respectively takes up a stream threshold of 5000 km$^2$ and 1000 km$^2$. It also explains the unexpectedly low $b$ values in deltas observed in **Fig. 3**. In contrast, the Arctic exhibits generally low uncertainty, likely because only one reference dataset is available above 60°N, reducing discrepancies and thus lowering remaining uncertainty.


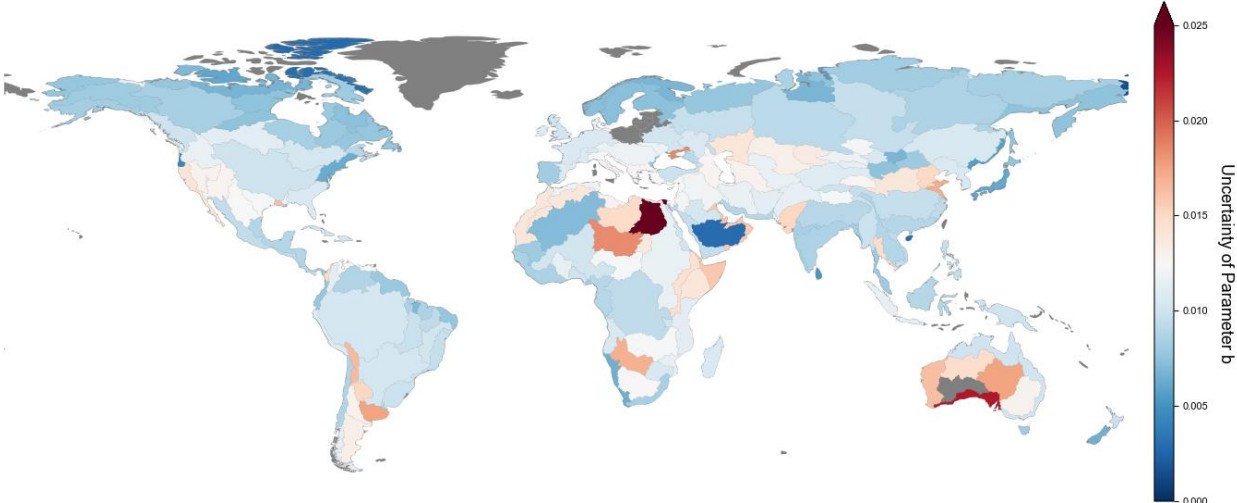

**Figure 9. Spatial pattern of the residual uncertainty of parameter *b* by basins. Residual** uncertainty is quantified as the standard deviation among all possible *b* values derived at different

percentiles (see Section 2.2 for details).

## 4.3 Floodplain definition and inundation maps

We also dedicate some discussions to the definition of floodplains here as numerous definitions exist for different intended uses. Geomorphically, a floodplain is an accumulation plain along a

watercourse, formed by unconsolidated sediment transported and deposited by the stream, usually flooded during high flows (Brierley and Fryirs, 2013). This definition emphasizes the formation process. From a hydrologist or a flood manager's perspective, the floodplain is often associated with inundation attached to certain flood strengths (Krizek et al., 2006), which can be also referred to as the hydraulic floodplain. Alternatively, focusing on material flux exchanges yields different

boundaries (Wohl, 2021). We consider these perspectives not contradictory but complementary in floodplain mapping processes as they highlight different aspects of floodplains. Specifically, geomorphic floodplains are predominantly shaped by low-probability but high-impact flood occurrences (Lindersson et al., 2021), which subsequently connects our goal of delineating a geomorphic floodplain with identifying a boundary that encompasses all potentially inundated

areas under extreme conditions. Therefore, we have used two 500-year return period flood maps as references for estimating our parameters, only to ensure a sufficiently large boundary for the

carrying out of this algorithm. This way, the geomorphic definition of a floodplain is still obeyed. While the FHG parameters can be approximated for various return periods (Nardi et al., 2006), our approach does not focus on nor involve a specific return period for inundation. In other words, our

goal is not to provide a mere substitute for inundation maps. Instead, we aim to leverage a river's geographical characteristics and hydrological extreme conditions, to identify scaling relationships that align with geomorphic principles, and to offer a more comprehensive understanding of global floodplain extents.

**4.4 Spatial scales of SHIFT**

The spatially varying parameters for SHIFT are derived at the scale of HydroBASINS Level-3 basins, which depicts 269 river basins globally with some containing aggregations of smaller basins. These aggregated basins are not hydrologically connected and are less suitable for our thresholding scheme that estimates one set of parameters for each basin, compared to the largest

basins which shares internally consistency hydrogeomorphic processes. A possible strategy to improve the scheme is to further divide these basins into smaller sub-basins, but smaller-scale analysis can increase the impact of reference data uncertainties especially in delta regions with high floodplain discordance (**Fig. 5a**). Parameters for Level-4 and Level-5 basins were also calculated (statistics are given in **Table S1**), but many basins had insufficient reference grids to

give reliable estimations. Considering the high data noise that may limit further integration of sub-basin level heterogeneity in estimating parameters, the spatial disaggregation scheme used by SHIFT (i.e., level-03) is sufficient in improving heterogeneity while offering reasonable physical interpretations of the parameters.

Lastly, when calculating HAND as the terrain attribute for SHIFT, we set an UPA threshold of 1000 km$^2$ to delineate the river network grids following past studies (Nardi et al., 2019; Rudari et al., 2015). A sensitivity test on a smaller threshold (50 km$^2$) not shown here suggests that more detailed floodplains around smaller rivers can be derived, but at the same time such a threshold can limit expected floodplains by large rivers. Conversely, a larger threshold, such as the 5000 km²

used by the JRC dataset, imposes a stricter criterion on river streams, leading to fewer river networks and reduced floodplain boundaries in areas like deltas. Thus, this study considers the

1000 km$^2$ UPA threshold to be valid. Future large-scale studies can further investigate the above-mentioned scale parameters, but we expect the gains to be minimal.


# 5   Conclusions

In this study, we develop an improved thresholding scheme for large-scale DEM-based floodplain delineation, the core of which being a stepwise estimation framework for Floodplain Hydraulic Geometry (FHG) parameters that respects the power law while better integrating spatial

heterogeneity from two publicly available hydrodynamic flood maps. We applied the framework at the scale equivalent to HydroBASINS Level-3 basins to derive localized FHG parameters as an update to previously global parameters that do not account for the heterogenous factors influencing floodplain extents. The optimized empirical exponent *b* in FHG exhibits statistically significant positive correlations with hydroclimatic conditions, particularly in major river basins. Based on

the proposed framework, we created a global geomorphic floodplain map named SHIFT (Spatial Heterogeneity Improved Floodplain by Terrain analysis) using terrain inputs from the 90-m MERIT-Hydro dataset, where SHIFT is demonstrated to capture both the global patterns and regional details of geomorphic floodplains well. The effectiveness of our framework is supported by:


1. The parameters show statistically significant but relatively weak relationships with hydroclimatic variables (e.g., AI, LAI), suggesting an enhanced representation of spatially heterogeneous hydrological and geomorphic information at the basin level.

2. The filtered data conforms to a relatively stable power law, suggesting a robust regionalized

scaling relationship.

3. Parameter changes lead to improved consistency with existing maps, with better differentiation between mainstreams and tributaries in major basins and more comprehensive representation of stream networks in aggregated river basins.

We provide the SHIFT data layers at two spatial resolutions (i.e., 90-m and 1-km) for the convenience of the users. The optimized parameters are also provided to support future studies.

Overall, we offer a framework for estimating spatially varying FHG parameters, contribute an updated geomorphic floodplain dataset, provide a better understanding of observable influences in the FHG scaling relationships, and expand on discussing the different focuses and implications of various floodplain mapping techniques. We hope our analysis to be helpful to enhance the understanding of current methodologies for defining and identifying active floodplains, especially in the context of changing climate.

## Author Contribution

Conceptualization: PL, KZ. Data curation: KZ, PL, ZY. Formal Analysis: KZ. Funding Acquisition: PL, KZ. Investigation: KZ, PL. Methodology: KZ, PL. Writing – original draft: KZ, PL. Writing – review & editing: KZ, PL, ZY.

## Data and Code Availability

SHIFT is openly available at https://zenodo.org/records/11835133 (Zheng et al., 2024). The core codes involved with terrain analysis and FHG parameter estimation is available at https://github.com/Mostaaaaa/SHIFT_floodplain.

## Conflict of Interests

The authors declare no conflict of interests.

## Acknowledgements

This study is supported by the Open Research Program of International Research Center of Big Data for Sustainable Development Goals, Grant No. CBAS2022ORP05, the National Natural Science Foundation of China (42371481), Yunnan Science and Technology Major Project (grant NO. 202302AO370012), and the Beijing Nova Program (20230484302). We acknowledge the funding support from the Fundamental Research Funds for the Central Universities, Peking

University on 'Numerical modelling and remote sensing of global river discharge' (no.
830    7100604136).

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
