# Peer review of "SHIFT: A DEM-Based Spatial Heterogeneity Improved Mapping of Global Geomorphic Floodplains"

_Earth System Science Data, 2023_

## Author Comment (AC3)

**Reviewer #3**

The paper is well-written and addresses a critical need of the community by developing a new, relatively finer resolution global scale floodplain map. It uses HAND as the driving topographic attribute. While the paper presents a comprehensive dataset, I do see some major conceptual limitations that make the dataset and the underlying logics questionable. Given these limitations and my strong reservations about ESSD's high standards with regards to study methods, I think this paper would be suitable for a regular hydrology or flood related journal.

Reply: Thank you for recognizing our work and bringing reasonable criticisms to our study, particularly concerning our conceptualization. Your concerns suggest that some potential confusion may need further clarification, which we believe can be addressed through improved framing and writing. We have significantly revised and restructured our Discussion section to clarify any potential confusion about concepts, and to provide a more objective presentation of our results and the contributions of our research. Below, you'll find our point-to-point replies and we hope they clears up your concerns.

(1) The purpose of topography-based hydrogeomorphic floodplain mapping is to (a) avoid complex and computationally intensive modeling approaches, and (b) map flood hazards without any specific return period of extreme event (eg 50, 100, 500 year flood). But this study overrides that concept and uses existing 500-year flood maps from two hydrodynamic models to calculate scaling parameters for HAND. Clearly, this opposes what we know about the science of hydrogeomorphic floodplain mapping. In short, the method proposed in this study takes years of development and conceptual knowledge in a confusing direction. If I have to use hydrodynamic models for creating a hydrogeomorphic model, then the whole idea of hydrogeomorphic modeling is meaningless.

Reply: Thank you for your comment on the possible confusion of our conceptualization. We would like to clarify that we are not using hydrodynamic models to create a hydrogeomorphic model but to better determine the floodplain boundary. Although the conceptual definition of a geomorphic floodplain does not involve such a boundary, in practice, it is essential that we obtain some sort of information, be it from hydrodynamic maps or in-situ measurements as the reviewer mentioned in the other comment, to help us define this boundary. By incorporating outputs from hydrodynamic maps (not models), we are obtaining Floodplain Hydraulic Geometry (FHG) parameters for the already established hydrogeomorphic modelling framework, and the maps from

hydrodynamic models have proven to be of use. The contribution in our study to the framework is to add information on spatial variability in parameters.

Conceptually, we believe that different definitions of floodplain boundaries are complementary rather than contradictory, each highlighting different facets of floodplain dynamics. In our case, the concept of a geomorphic floodplain emphasizes the formation process of floodplains, but it is also predominantly shaped by low-probability, high-impact flood occurrences (Lindersson et al., 2021). Considering that FHG describes the extent of inundation depth (hydrological factor) with drainage area (geomorphic factor), our goal of delineating a geomorphic floodplain is subsequently connected with identifying a boundary that encompasses all potentially inundated areas under extreme conditions. **Therefore, we've used two 500-year return period flood inundation maps as references for estimating our parameters, only to ensure a sufficiently large boundary for the carrying out of this algorithm**. **This way, we believe that the geomorphic definition of a floodplain is still obeyed.** While the FHG parameters can be approximated for various return periods (Nardi et al., 2006) and can subsequently be viewed from an inundation perspective, our approach does not focus on a specific return period for inundation. In other words, our goal is not to provide a mere substitute for inundation maps; rather, we aim to consider both the stream's geographical characteristics and hydrological extreme conditions, to identify scaling relationships that align with geomorphic principles, and to offer a more comprehensive understanding of floodplain dynamics.

Thank you again for pointing out this potential confusion on our conceptualization. To better address your concerns, we have largely revised the above discussion on floodplain boundary definition and delineation in the revised Section 4.3 in Discussion. Additionally, to facilitate future studies and reduce computational efforts, we will provide our spatially varying parameters for easier application. These parameters are now available at the same Zenodo repository at https://zenodo.org/records/10440609.

(2) Alongside the conceptual limitation, the work is self-contradictory. The authors on and on tag their approach as parsimonious and existing hydrodynamic models as uncertain (see Lines 84-86). Parameterizing HAND with two hydrodynamic model-based flood maps, as the authors did, is in no way a parsimonious method. This is also not a practical method. Because if I don't have hydrodynamic models existing in my area of interest (let's forget about uncertainty for the sake of discussion), I won't be able to reproduce the authors' method.

Reply: We respectfully disagree with this point and would like to emphasize that our approach remains parsimonious. Strictly speaking, we did not use hydrodynamic

models but rather publicly available flood maps as references, despite their uncertainties and inconsistencies. Thus, no complex models or simulations are involved in our method, as the core process is described by a power-law. The most intricate part of our study is the data filtering scheme, but it still demands significantly less computational effort compared to hydrodynamic models. Besides, we believe that the issue of parsimony and practicality can be better addressed by providing our optimized scaling law parameters. For anyone who wish to reproduce the method/results using terrain data, it is easy to grab our results and derive new maps of their own, thus replicating our method should be feasible.

In addition to the above, in this revision, we have carefully conducted additional investigations into the concern on the uncertainty related to using hydrodynamic maps. Despite their acknowledged inconsistencies, the reference maps we used are informed by climatic forcing and are subsequently expected to offer a more spatially heterogeneous basis than universal geomorphic parameters. In other words, while we do acknowledge these maps can be uncertain, they contain useful information that can be applied to constrain geomorphic floodplain boundaries. We have thus introduced a rigorous data filtering process to optimize the parameters best conforming to the power law contained within the data. Our results show that the filtered data conform well to the power law (see revised Figure 9), supporting the validity of our approach.

We hope this resolves the "self-contradictory" concern for our work. Revisions have also been made more clearly to address your conceptual concerns: for detailed explanations of using these maps as references, please see the newly added Section 4.1, and for the remaining uncertainty please refer to Section 4.2. Our supplied parameter maps can be found in the zenodo repository for more expert users.

Many examples of HAND's parsimonious applications already exist in literature. HAND is parsimonious in operationalized flood prediction systems where a streamflow or stage height (the H in authors' scaling equation) comes from an operational watershed hydrology simulation model followed by a process of automatic synthetic rating curve generation. See examples like https://doi.org/10.31223/osf.io/hqpzg

Reply: We thank the reviewer for this comment, and we are actually aware of the alternative thresholding methods for HAND that are available and widely utilized. The paper you provided outlines two approaches: 1) directly estimating stage height, which is useful when in-situ measurements are available, and 2) using a synthetic rating curve, as also calculated from terrain-based methods. The latter method is indeed effective for large-scale applications and is used by the US National Water Model, but it introduces additional sources of uncertainty as it requires estimated Manning's

coefficients for water-stage estimation and it is also computationally very demanding as it has not been accomplished worldwide. Therefore, outside of the United States where high-quality data is available, replicating this globally poses significant challenges.

To compare with, the FHG method requires only terrain input, which is recognized as the least uncertain component in global floodplain mapping when using hydrodynamic models. The necessary information is encapsulated in the parameters, making it easier to identify the influence of each parameter. Therefore, we consider the FHG thresholding approach to be more globally consistent and easily applicable, and still a useful contribution to the community. We have added an additional section on FHG in Section 4.1 and included a paragraph on other thresholding schemes for HAND to address your concerns. We have also supplied parameters for use by future researchers to make it parsimonious.

(3) The aridity came out of nowhere. I think bringing aridity into the mix was arbitrary and unnecessary.

Reply: Thank you for pointing out this potential confusion, which was also brought up by other reviewers and which we have carefully addressed in this revision. We'd like to clarify that including aridity in our analysis was purposeful and based on our hypothesis. Our estimated parameters aim to capture spatial heterogeneity of geomorphic floodplain forming factors and, if possible, we should be able to identify significant relationship between examined factors and our derived parameters. Due to uncertainties with the data as well as the scaling law itself, we do not expect the relationship to be perfect. We hypothesized that in humid areas, the stronger discrepancy between small and large rivers would lead to a stronger dominance by larger rivers. While the correlation with the Aridity Index (AI) is not strong, its significance supports our parameter estimation efforts. The largest basins, being hydrologically connected and thus internally consistent in hydrological characteristics, result in stronger correlations with these factors as expected. Despite the seemingly loose correlations, our analyses may still be helpful in identifying geomorphic floodplain-forming mechanisms.

To address your comment regarding the clarity of our purpose, we have clarified our hypothesis and strengthened the tests we conducted in this revision. In terms of writing, we have explained this in the Methods section and elaborated on it in the revised Section 4.1 on our hypothesis with the FHG parameters.

Experimentally, we conducted two additional analyses. First, we included more

variables in our analysis, such as LAI, terrain (mean and deviation), and soil factors (soil components in a river buffer). Our hypothesis was that AI would be the most significant factor, with LAI inherently related to AI, while terrain and soil might also be related but with less clear mechanisms. The results showed that AI was indeed the most significant, with LAI only significant in large basins. Other factors exhibit inconsistent correlations with b, also as expected. We also tested the estimation of the parameters at different scales (i.e., Level-4 and Level-5 basins) to increase the sample size. The results showed that AI and LAI have statistically significant relationships with the exponent b, while terrain factors showed significant but much weaker relationships, followed by soil factors that do not show statistically significant relationships with b (see our Supplementary Figure 1, included below).

While the correlations shown in the above analyses may not be very strong, they meet our expectations: AI is significant as the primary factor for explaining the spatial variability of b, LAI plays a role, and terrain might be related but not showing readily detectable correlations with the exponent b.

**Table 1 (Supplementary Table 1 in the revised manuscript). Correlation of FHG parameter $b$ and relevant hydroclimatic factors.** Results from Level-4 and Level-5 basins are filtered by the amount of available reference grids in the basin. Soil data are from the Soilgrids 2.0 dataset [3] and processed within a 10-km buffer calculated by hydrological distance.

| | | Aridity Index | LAI | Elevation Mean | Elevation STD | Clay | Silt | Sand |
|---|---|---|---|---|---|---|---|---|
| **Level-3** | All | 0.335*** | 0.083 | -0.007 | 0.121 | 0.152* | 0.170* | -0.041 |
| | Largest | 0.680*** | 0.668*** | -0.165 | 0.208 | 0.314 | -0.134 | -0.042 |
| **Level-4** | | 0.338*** | 0.256*** | 0.131** | 0.246*** | -0.067 | 0.050 | -0.003 |
| **Level-5** | | 0.405*** | 0.349*** | 0.104*** | 0.188*** | -0.033 | -0.019 | 0.033 |

We've revised our manuscript accordingly to include both the more clearly stated hypothesis and our interpretations. Please refer to the newly performed analyses in Supplementary Table 1, and more objective statements of our parameters and hypothesis in Section 4.1.

References:

1. Lindersson, S., Brandimarte, L., Mård, J., and Di Baldassarre, G.: Global riverine flood risk – how do hydrogeomorphic floodplain maps compare to flood hazard maps?, Nat. Hazards Earth Syst. Sci., 21, 2921–2948, https://doi.org/10.5194/nhess-21-2921-2021, 2021.

2. Nardi, F., Vivoni, E. R., and Grimaldi, S.: Investigating a floodplain scaling relation using a hydrogeomorphic delineation method: HYDROGEOMORPHIC FLOODPLAIN DELINEATION METHOD, Water Resour. Res., 42, https://doi.org/10.1029/2005WR004155, 2006.

3. Poggio, L., de Sousa, L. M., Batjes, N. H., Heuvelink, G. B. M., Kempen, B., Ribeiro, E., and Rossiter, D.: SoilGrids 2.0: producing soil information for the globe with quantified spatial uncertainty, SOIL, 7, 217–240, https://doi.org/10.5194/soil-7-217-2021, 2021.

---

## Author Response (AR1)

**Author's Comment**

We have carefully addressed the comments and suggestions provided by the reviewers, resulting in significant improvements to our manuscript. Below, we summarize the major changes made, organized by the issues addressed, including the corresponding responses to the reviewers and the sections of the manuscript where these changes are reflected.

1. **FHG parameters**. We clarify and strengthen the hypothesis regarding the relationship between the Floodplain Hydraulic Geometry (FHG) parameters and hydroclimatic conditions. Specifically:

   1. **Hypothesis Articulation:** We consider aridity to be the primary determining factor due to the assumption that in a humid basin, rivers with larger upstream drainage areas (UPA) would have greater dominance over smaller river segments. Vegetation and other factors such as terrain and soil composition might also influence the results.

   2. **Correlation Testing:** We tested the correlation with factors including Leaf Area Index (LAI), mean elevation, elevation standard deviation, and soil components (clay, silt, and sand). Only the Aridity Index showed significant correlations with our parameters, while LAI showed significant correlations in the largest basins. Other factors did not present uniform results.

   3. **Scale Testing:** We tested our hypothesis on different scales (Level-4 and Level-5 basins), finding that both the Aridity Index and LAI exhibit significant correlations, suggesting these factors might be influential.

   4. **Parameter Provision:** We provided our parameters to help derive regional datasets or apply hydrogeomorphic methods to different terrain data or resolutions, available at our Zenodo repository.

      **Sections Updated:** Sections 4.1 and 4.4, with Supplementary Table 1.

2. Assessment of Accuracy and Effectiveness of the Methods.

   1. **Comparison with Universal Parameters:** We added a comparison with MERIT-Hydro using universal parameters (UP) for parameter estimation. This analysis showed terrain differences impact model agreement and that our locally optimized parameters offer better

performance in most cases.

2. **Clarification of Effectiveness:** We clarified that the effectiveness is supported by the parameters' statistically significant relationships with variables like the Aridity Index, conformity to a stable power law, and improved consistency observed in spatial patterns.

    **Sections Updated:** Section 3.3.

3. Conceptual Improvements.

    1. **Definition of Floodplain Boundary:** We clarified that our study focuses on identifying boundaries encompassing all potentially inundated areas under extreme conditions using two 500-year return period flood maps.

    2. **Distinction and Application:** We emphasized that we are not using hydrodynamic models to create a hydrogeomorphic model but to obtain locally varying FHG parameters. Our approach aims to balance geomorphic principles with hydrological aspects.

        **Sections Updated:** Section 4.3.

4. Improvements on our technical flow.

    1. **Parameter 'b' Estimation:** We modified the technical details of parameter 'b' estimation, specifically the binning parameter, adding a constraining mechanism to handle data noise. This resulted in stabler estimates for large basins and a clearer pattern of global residual uncertainty.

    2. **Target Function for Parameter 'a':** We changed our target function to balance information from both datasets, using Fleiss's Kappa (FK) and a penalty term to reduce bias.

    3. **Uncertainty Representation:** We defined our metric as residual uncertainty, assessing how well data conforms to the power law, aiming to provide a clearer explanation of the uncertainty and a quantitative assessment of our approach.

        **Sections Updated:** Section 2.2 for methodological details and Section 4.2 for uncertainty discussion.

Other revisions include but are not limited to include the change of palettes, clarification of technical details. Please find our point-to-point response below.

**Reviewer #1**

The authors have developed a global geomorphic model of fluvial floodplains, notably at 90 m resolution. The authors made use of global elevation, flow direction, and drainage area models along with the global HydroBASINS boundaries for their analysis.

The methods presented here closely follow the methods of Nardi et al., 2019 (i.e., GFPlain250) which uses height above nearest drainage (HAND) with a floodplain hydraulic geometry (FHG) thresholding scheme. FHG suggests that potential inundation depth can be represented as a function of a river's upstream draining area. Here, the authors argue that FHG parameters optimized for each basin, as opposed to global values, will better represent the spatial heterogeneity of global basins and ultimately be of benefit to floodplain delineation.

The authors propose an iterative process with starting values based on previous knowledge that converges on suitable parameter values for each global basin. The authors found that Parameter $b$ in the FHG model loosely corelates with a basin's aridity index. Finally, the authors use two global hydrodynamic inundation maps (JRC and GAR) and another geomorphic floodplain model (GFPlain250) as reference data for comparison.

The authors have a logical claim; basins across the world are heterogenous and locally optimized FHG parameters could produce better models of floodplains when compared to global parameters. Of note, Nardi et al., justified the use of global coefficients by finding reasonable measure-of-fit values with varying $b$ parameters. However, they also supported the notion that regional values for the scaling law parameterization could be further refined to capture local climatic variations.

SHIFT data and code were easily accessible. In North America, SHIFT aligns well with GFPlain with some noticeable differences. Specifically, in North America, SHIFT tends to estimate a narrower floodplain in comparison to GFPlain. Both products have notable examples of areas identified as floodplains that are omitted by the other.

I have several comments I would encourage the authors to consider.

**Reply**: Thanks for your clear and comprehensive comment. You'll find our point-to-point replies below.

• 170 - I'm unclear on why 34 'major' river basins were selected for further analysis. The authors rely on the results in these 34 basins as evidence throughout their paper. Please explain the selection of these basins, what is significant about them, and what is the justification for analyzing them independently.

**Reply**: Thank you for pointing out this. The selection of the major river basins was based on the largest river basins aggregated by MERIT-Basin that are hydrologically connected. Specifically, we traced from all the outlet basins inland (and aggregated the inland rivers accordingly) to identify the largest basins. We performed a separate investigation into the floodplain hydraulic scaling relationship for major river basins, based on the assumption that larger river basins may exhibit more consistent floodplain hydrological processes and formation mechanisms, thus having stronger scaling relationships. This consistency may also provide a clearer context to investigate the relationships between relevant factors and the scaling exponent. However, some of the Arctic basins fall beyond the boundary of our reference datasets, so we manually excluded those basins. In this revision, we have refined our selection process by detecting whether the centroid of a basin is within the Arctic to make the definition clearer. As a result, we have identified 7 Arctic basins to exclude, leaving 33 major river basins for our analysis. We have added the necessary explanations to our revised manuscript to clarify the selection criteria and the justification for analyzing these basins independently.

**Revision:**

(Line 185, Section 2.1) "Among the Level-3 basins, the 40 largest hydrologically connected basins were manually selected, based on the hypothesis that connected basins better apply the scaling law due to shared attributes within the same hydrological system. Seven of these 40 basins, with centroid located above 60°N, were excluded since one of our reference maps does not cover regions above 60°N."

- 233 – This could use more explanation. Why did the authors choose to define river as a function of UPA versus using the delineated river network in MERIT Hydro? Why select 1000 km2? The authors touch on this at the end of the paper.

**Reply**: The MERIT-Hydro dataset does not originally provide vector-based river network datasets [1], but MERIT-Basins does [2]. MERIT-Basins is delineated by setting a threshold on the UPA of the river, but it uses a 25 km² threshold, which would include too much small streams for global floodplain delineation. Conversely, a larger threshold, such as the 5000 km² used by the JRC dataset, imposes a stricter criterion on river streams, leading to fewer river networks and reduced floodplain boundaries in areas like deltas or other medium- to large-size rivers. We chose to define rivers as a function of UPA with a threshold of 1000 km² because it balances how many rivers are incorporated in the global-scale

floodplain delineation process. This threshold ensures significant river networks are included without overwhelming the analysis with smaller, less relevant streams, which was also adopted by Nardi et al. (2019). Additionally, while MERIT-Hydro provides Height Above Nearest Drainage (HAND) data, it is based on a 0.5 $km^2$ threshold, which contains too many small streams not relevant for our purposes. Thus, we have chosen to use the 1000 $km^2$ and we have expanded our explanation in the methods section (2.2) and provided relevant discussion in section 4.4 to clarify this choice.

**Revision:**

(Line 185, Section 2.2) "River grid here is identified by applying a 1000 $km^2$ threshold to the Upstream Drainage Area (UPA), supported by previous studies (Nardi et al., 2019). The threshold is determined by preliminary experiments to ensure that it is neither too small, which would misattribute large-river-dominated floodplains to small rivers, nor too large, which would overlook rivers with notable influence."

(Line 766, Section 4.4) "Lastly, when calculating HAND as the terrain attribute for SHIFT, we set an UPA threshold of 1000 $km^2$ to delineate the river network grids following past studies (Nardi et al., 2019; Rudari et al., 2015). A sensitivity test on a smaller threshold (50 $km^2$) not shown here suggests that more detailed floodplains around smaller rivers can be derived, but at the same time such a threshold can limit expected floodplains by large rivers. Conversely, a larger threshold, such as the 5000 $km^2$ used by the JRC dataset, imposes a stricter criterion on river streams, leading to fewer river networks and reduced floodplain boundaries in areas like deltas. Thus, this study considers the 1000 $km^2$ UPA threshold to be valid. Future large-scale studies can further investigate the above-mentioned scale parameters, but we expect the gains to be minimal."

- 350 – What method was used to resample to 1-km?

**Reply**: For continuous data like UPA and HAND, we used median as the resampling method. As for categorical data like the reference datasets and watershed division, we used mode as it accounts for the majority of information in the selected area. The details are added to the corresponding section.

**Revision:**

(Line 409, Section 2.2) "We used median as the resampling method for continuous variables like UPA and HAND, and mode for categorical data, such as the reference maps, SHIFT, and watershed division."

- 352 – I see many permanent water bodies in the final SHIFT product. (e.g., the North American Great Lakes).

  Reply: In the previous version of our 1-km product, we've marked permanent water bodies with identifier 2, which can be easily filtered by putting a mask. We've also put the identifier for water bodies in the version 2 of our updated data. We will put our revised data in the Zenodo repository.

- 450 - Use of overall accuracy overly rewards correctly classifying the 94.5% (author's estimates) of the world's land area that is not a floodplain. I would be more persuaded by overall accuracy if the authors were to limit their accuracy analysis to some reasonable distance from your river network (e.g., 1km, 10km).

  Reply: Thank you for bringing up this good point and we agree with your suggestion. To assess the consistency of our data with two other maps, in this revision, we've changed it to MAI because OA will overly reward non-floodplain areas as the reviewer noticed (Fig. 7). For the pairwise comparison, we've added the OA within a buffer of 20-km in the pairwise comparison section to provide a better understanding. The buffer here is calculated by the hydrological distance, that is by d-8 flow direction, and we've tried different buffer threshold from 5-km to 50-km. Overall, the OA statistics shown by all buffered threshold showed quite similar patterns, and since 5-km to 10-km may be too small since it'll be continuously 10-km of floodplain in some large basins, so a 20-km buffer is finally decided. The buffered patterns generally align with the un-buffered OA because OA considers non-floodplain areas, and the buffer merely adjusts the extent of these areas considered. In contrast, MAI focuses exclusively on overlapping floodplain areas and does not consider non-floodplain areas, resulting in different patterns. Results and descriptions are revised correspondingly.

  Revision:

  (Line 425, Section 2.2) "The two types of indices applied here have different focuses: OA considers non-floodplain areas, while MAI focuses exclusively on overlapping floodplain areas. Considering the overall landmass is non-floodplain, we also calculated OA within 20-km buffer zones, with distance measured as the hydrological distance to the stream."

[Figure]

**Fig. 1 (Fig. 7 in the revised manuscript). Validation of SHIFT against two reference datasets.** In the bivariate map, the two variables are the MAI against the JRC map (magenta) and the GAR map (yellow). A balanced MAI results in red basins.

- 454 – I would think to prove "the effectiveness of our parameter estimation scheme in capturing information from the reference maps", I would need to see this same accuracy measurements but with global values used (e.g., the Ndari et. al., values: *b*= 0.3, *a* = 0.01) and the deltas.

> **Reply**: Thank you for your suggestion. It is indeed a great idea to include this comparison. We have now added a comparison with MERIT-Hydro using universal parameters (UP) in this revision. Statistically, the total area of UP is 50.85% larger than SHIFT. We further conducted a pairwise comparison including UP, and the spatial distribution is shown in Fig. 8b (shown below). Among all pairs, SHIFT-JRC performed best in 62 basins, SHIFT-GAR in 74, UP-JRC in 8, and UP-GAR in 37. This result offers evidence that the derived parameters are effective in deriving better floodplain maps.

[Figure]

**Fig. 2 (Fig. 8b in the revised manuscript).** Bivariate choropleth map of the highestperformance MAI pair among four pairs (SHIFT & JRC, SHIFT & GAR, UP & JRC, UP & GAR) and the corresponding MAI value for each basin Different pairs are represented by different hues, with higher MAI values shown in higher saturation. Basins where a SHIFT-pair performs best are marked in cool colors, while those where a UP-pair performs best are represented in warm colors.

In the boxplot showcasing consistency between the hydrodynamic maps and the geomorphic maps, however, we observed vast statistical difference between GFPlain and UP larger than that of SHIFT and UP. The explanation for these observations is twofold. First, the terrain inputs of GFPlain and UP are different. MERIT-Hydro includes hydrological corrections where all water body values are manually lowered, resulting in significantly lower HAND values and subsequently larger inundation extents under the same parameter applied. Second, the difference in SHIFT and UP is underrepresented in the boxplot, as SHIFT-JRC pair usually has high consistency where UP-GAR agrees better, and vice versa. We have documented these interpretations objectively in the revised texts.

[Figure]

**Fig. 3 (Fig. 8a in the revised manuscript).** Boxplots of pairwise analysis among SHIFT, GFPlain, UP (MERIT-Hydro but with Universal Parameters), JRC and GAR across three metrics: MAI (left), OA (middle) and OA within a 20-km buffer (right). Two group comparisons are marked in different colors (magenta for JRC and yellow for GAR). Statistics for all basins with valid data inputs (see Methods) are shown in blue boxes, and those for the 33 major river basins are shown in orange.

**Revision:**

(Line 413, Section 2.2) "After getting the updated floodplain boundary with the optimized parameters (SHIFT), we conduct a pairwise consistency analysis among five maps, i.e., SHIFT, GFPlain250m, UP (Universal Parameters, applying

*b* = 0.3 and *a* = 0.01 on MERIT-Hydro), JRC and GAR. UP was generated to allow the assessment of how changes in parameters influence the results."

(Line 585, Section 3.3) "To better understand the impact of our estimated parameters on the consistency performance, we analyze the most consistent pair and corresponding MAI values for each basin. Among all pairs, SHIFT-JRC aligns the best in 62 basins, with SHIFT-GAR in 74, UP-JRC in 8, and UP-GAR in 37 (**Fig. 8b**). This validates that SHIFT exhibits better consistencies with the reference maps even though the difference between SHIFT and UP seems not statistically significant (**Fig. 8a**). Spatial patterns (**Fig. 8b**) show that SHIFT-JRC pairs aligns best in humid major basins (e.g., the Mississippi and Amazon) and very arid regions (e.g., the Taklamakan and central Australia). SHIFT-GAR pairs are the most consistent in mountainous regions (e.g., the Rockies and Andes), aggregated deltas (e.g., eastern Australia and southern Africa), islands (e.g., Indonesia), and inland river basins (e.g., the Tibetan Plateau) where few rivers meet the 5000 km² drainage area threshold of JRC. In contrast, cases where UP pairs align best are less common. UP aligns better with GAR due to their shared large prediction extents, such as around the Caspian Sea. In rare instances where UP-JRC pairs perform best, it is typically in deltas or regions where SHIFT-GAR performs well, such as deltas and islands. This is likely because our method balances consistency between the datasets, but GAR's wider prediction coverage makes this strategy less effective in these infrequent cases.

Note that GFPlain and UP use the same parameter for its geomorphic delineation, but their consistency with JRC and GAR differs significantly (**Fig. 8a**). This is because GFPlain uses 250-m SRTM as the terrain input, while UP uses MERIT-Hydro, which has undergone hydrological correction to lower the elevation of waterbody pixels, resulting in higher HAND values and smaller floodplain extents. GAR, which generally overpredicts floodplain extents especially in arid regions, aligns better with GFPlain. The overprediction of GAR is evidenced by GAR-pairs having the lowest OA, as OA strictly penalizes overprediction. At the same time, we found the difference between SHIFT and UP may be under-represented in the statistical plots (**Fig. 8a**) while the actual impact of variable parameters brought by SHIFT is substantial: the global floodplain extent estimates are 14.95 million km² for UP and 9.91 million km² for SHIFT, showing a 50.85% difference in total predicted areas. Additionally, regions where UP-GAR has the highest consistency (**Fig. 8b**) generally coincide with regions where SHIFT-JRC aligns best. This reversed pattern of consistency further supports that the statistical differences

between UP and SHIFT are underrepresented in **Fig. 8a**.”

- 472 – “Superiority” is an overstatement. Agreement does not equate to superiority.

  **Reply**: We agree and this was removed in this revision. We stepped back and more objectively mentioned the improvement of agreement of SHIFT.

  **Revision:**

  (Line 28, Abstract) “Our results demonstrate that SHIFT validates better with reference maps than both hydrodynamic modeling and DEM-based approaches with universal parameters. The improved delineation is mainly with better differentiation between mainstreams and tributaries in major basins and a more comprehensive representation of stream networks in aggregated river basins.”

  (Line 796, Section 5) "Parameter changes lead to improved consistency with existing maps, with better differentiation between mainstreams and tributaries in major basins and more comprehensive representation of stream networks in aggregated river basins."

- 560 – I'm not sure I would call FHG correlation with hydroclimatic conditions 'reasonable'. There is a loose correlation. Earlier the authors described it as 'statistically significant but weak'. That is a more apt description.

  **Reply**: Thank you for raising this reasonable concern. We agree that "significant but weak" is a more accurate description. By "reasonable", we meant that the correlation is within our expectations, as it is important to notice that we did not anticipate a perfect correlation between exponent 'b' and hydroclimatic conditions for several reasons. First, the scaling relationship is a simplified theory that summarizes floodplain-forming processes, and the strength of this relationship itself warrants further investigation. Second, there are inherent noises in the two maps (as inherited from their own model chain errors), which can also limit the strength of the observed relationship. Since 'b' is an empirical parameter, and its physical interpretation remains an area of study [4], our goal was to identify any observable patterns between our optimized b with other factors, rather than expect perfect correlations.

  We'd like to argue though, that while the correlation with the Aridity Index is not strong, its statistical significance supports the effectiveness of our parameter estimation methods, which helps to derive spatially-varying parameters that are physically meaningful. The largest basins, being hydrologically connected and are internally consistent in hydrological characteristics, also result in stronger

correlations with the factors as expected.

To address our focus on interpreting the exponent b and its linkage with physical factors, we conducted two additional experiments in the hope of more comprehensively identifying factors that can be related to the geomorphic floodplain-forming processes:

    i.   **Additional Variables**: We included more variables in our analysis, such as LAI, terrain (mean and deviation), and soil factors (soil components in a river buffer). Our hypothesis was that AI would be the most significant factor, with LAI inherently related to AI, while terrain and soil might also be related but with less clear mechanisms. The results showed that AI was indeed the most significant, with LAI only significant in large basins. Other factors exhibit inconsistent correlations with b, also as expected.

    ii.   **Different Scales**: We also tested the estimation of the parameters at different scales (i.e., Level-4 and Level-5 basins) to increase the sample size. The results showed that AI and LAI have statistically significant relationships with the exponent b, while terrain factors showed significant but much weaker relationships, followed by soil factors that do not show statistically significant relationships with b.

While the correlations shown in the above analyses may not be very strong, they meet our expectations: AI is significant as the primary factor for explaining the spatial variability of b, LAI plays a role, and terrain might be related but not showing readily detectable correlations with the exponent b. We've revised our manuscript accordingly. Please refer to the newly performed analyses in Supplementary Table 1, and more objective statements of our parameters and hypothesis in Section 4.1.

**Revision:**

(Line 21, Abstract) "The estimated FHG exponent exhibits a significant positive relationship with the basins' hydroclimatic conditions, particularly in 33 of the world's major river basins, indicating the ability of the approach to capture fingerprints from heterogeneous hydrological and geomorphic influences."

(Line 783, Section 5) "The optimized empirical exponent *b* in FHG exhibits statistically significant positive correlations with hydroclimatic conditions, particularly in major river basins."

- 561 – I'm not convinced this loose correlation proves effectiveness of the methods.

    **Reply**: Thank you for your comment. We agree that it is challenging to prove the

effectiveness of our methods without ground truth. However, we'd like to argue that the correlation analysis is only an indirect way of demonstrating that it is meaningful to derive spatially-varying exponent b for floodplain delineation, because these spatially-varying parameters are related to possible floodplain-forming factors, but not random values leading to floodplain maps matching with the hydrodynamic model outputs. To be more specific, we believe the effectiveness is indirectly supported by: 1) the parameters have weak but statistically significant relationships with variables like the Aridity Index, which aligns with our hypothesis; 2) the filtered data conforms to a relatively stable power law, indicating a relatively robust scaling relationship, and 3) changes in parameters result in improved consistency, and they are observed in spatial patterns. For a detailed explanation of our hypothesis and the correlation, please refer to Section 4.1 and our previous reply. For the spatial pattern of improvement, please see our subsequent reply.

**Revision:**

(Line 662, Section 4.1) "Second, our estimated parameters aim to capture fingerprints from spatially varying hydrological and geomorphic processes that can influence the floodplain extent. We consider aridity as the primary factor influencing the spatial variability of $b$, based on the assumption that in humid basins, rivers with larger upstream drainage areas exert greater dominance over smaller segments in shaping floodplains. Vegetation also plays a role, as it influences runoff generation and modulates soil erosion, both key to floodplain formation. Additionally, factors such as terrain and soil composition might influence the results. Given the data uncertainties and the complex physical interpretations of $b$, it is important to note that we do not expect perfect relationships between these factors and the derived exponent $b$. The correlation analysis indeed aligns with our expectations: AI is statistically significant in explaining the spatial variability of $b$, while LAI plays a role and terrain does not show strong correlations with $b$. Soil compositions (Poggio et al., 2021) do not exhibit a consistent pattern across analyses done at different scales (**Table S1**). Despite the not-so-strong correlation with AI and LAI, its statistical significance supports the effectiveness of our proposed methods, which helps to derive spatially-varying parameters that are also physically meaningful."

(Line 791, Section 5) "The parameters show statistically significant but relatively weak relationships with hydroclimatic variables (e.g., AI, LAI), suggesting an enhanced representation of spatially heterogeneous hydrological and geomorphic

information at the basin level."

- 566 – I'm not convinced of "superior consistency". Sometimes SHIFT is part of the highest agreement pair in a basin and sometimes it is not (Fig 7). The authors mention "superior consistencies" in the abstract as well. I'm not sure how to interpret that phrase.

**Reply**: We agree and this was removed in this revision. We stepped back and more objectively mentioned where improvements occur and what leads to the change of consistency. Descriptions are now better delivered in section 3.3.

**Revision:**

(Line 30, Abstract) "The improved delineation is mainly with better differentiation between mainstreams and tributaries in major basins and a more comprehensive representation of stream networks in aggregated river basins."

(Line 571, Section 3.3) "Prominently, it shows that the consistency between SHIFT and JRC significantly improves over UP and GFPlain, but that with GAR does not (as shown in MAI). The consistency pattern can be explained by delving into the inner working of each dataset. For large basins, SHIFT highlights the mainstreams and reduces prediction of tributaries, thus aligning more closely with JRC as it highlights major rivers, leading to a decrease in consistency with GAR. UP and GFPlain align better with GAR in these regions, as they all tend to overpredict, especially in tributaries. For other basins, SHIFT strikes a balance between the two datasets. Comparing SHIFT with UP, SHIFT increases the lower interquartile range for JRC's OA and the upper interquartile range for GAR's OA, highlighting a general improvement with SHIFT. For MAI, the upper quartile with GAR has decreased while the lower quartile has improved, suggesting a consistency trade-off between the two datasets. Notably, all geomorphic maps show a better consistency with the hydrodynamic outputs than the hydrodynamic pair, proving again that the hydrogeomorphic delineation method is a more globally consistent framework.

To better understand the impact of our estimated parameters on the consistency performance, we analyze the most consistent pair and corresponding MAI values for each basin. Among all pairs, SHIFT-JRC aligns the best in 62 basins, with SHIFT-GAR in 74, UP-JRC in 8, and UP-GAR in 37 (Fig. 8b). This validates that SHIFT exhibits better consistencies with the reference maps even though the difference between SHIFT and UP seems not statistically significant (Fig. 8a). Spatial patterns (Fig. 8b) show that SHIFT-JRC pairs aligns best in humid major basins (e.g., the Mississippi and Amazon) and very arid regions (e.g., the

Taklamakan and central Australia). SHIFT-GAR pairs are the most consistent in mountainous regions (e.g., the Rockies and Andes), aggregated deltas (e.g., eastern Australia and southern Africa), islands (e.g., Indonesia), and inland river basins (e.g., the Tibetan Plateau) where few rivers meet the 5000 km² drainage area threshold of JRC. In contrast, cases where UP pairs align best are less common. UP aligns better with GAR due to their shared large prediction extents, such as around the Caspian Sea. In rare instances where UP-JRC pairs perform best, it is typically in deltas or regions where SHIFT-GAR performs well, such as deltas and islands. This is likely because our method balances consistency between the datasets, but GAR's wider prediction coverage makes this strategy less effective in these infrequent cases."

- Fig 7 – It looks like GFPlain has higher agreement with GAR and SHIFT has higher agreement with JRC. Any explanations as to why this is?

  **Reply**: Thank you for pointing that out. Indeed, GFPlain has higher agreement with GAR, while SHIFT shows higher agreement with JRC. In our original manuscript, we suggested that this might be because GAR tends to overpredict in certain regions such as in dry regions, similar to GFPlain. Now that we've performed the analysis with UP (i.e., universal parameters), we also observed that UP tends to provide larger estimates in some areas, whereas the estimates of MERIT-Hydro are inherently smaller. As seen in Figure 8b, although taking up a small percentage, UP still aligns better with GAR as both of them tend to over-predict floodplains, such as areas around the Caspian Sea. SHIFT aligns well with JRC as it highlights the inundation of large rivers, especially in the major river basins.

   **Revision:**

  (Line 603, Section 3.3) "This is because GFPlain uses 250-m SRTM as the terrain input, while UP uses MERIT-Hydro, which has undergone hydrological correction to lower the elevation of waterbody pixels, resulting in higher HAND values and smaller floodplain extents. GAR, which generally overpredicts floodplain extents especially in arid regions, aligns better with GFPlain. The overprediction of GAR is evidenced by GAR-pairs having the lowest OA, as OA strictly penalizes overprediction."

- Why include JRC & GAR and SHIFT & GFPlain combinations in the choropleth map? I'm less interested in where the two hydrodynamic models (JRC & GAR) or the two geomorphic models (GFPlain & SHIFT) agree and I'm more interested in where

SHIFT outperforms or underperforms against GFPlain. That is, where does GFPlain better align with hydrodynamic models and where does SHIFT better align with hydrodynamic models?

> **Reply**: Thank you for your suggestion. We have redesigned our experiments in the pairwise comparison accordingly. Now, group comparisons were conducted with JRC and GAR, where each hydrodynamic map was tested against SHIFT, GFPlain, and UP (Universal Parameters on MERIT-Hydro, see above). The JRC-GAR pair serves as the baseline. For the choropleth map, we only compared SHIFT and UP to the two hydrodynamic maps individually to see which pair performs best and to identify any patterns in their performance. GFPlain is excluded as to address your comments on showing results in localized parameters. These results show that SHIFT outperforms UP for majority areas in Fig. 8a (see blue and green areas where they have the highest MAI), which is a proof that the estimated parameters of SHIFT are useful.

> **Revision:**
>
> (Line 589, Section 3.3) "Spatial patterns (Fig. 8b) show that SHIFT-JRC pairs aligns best in humid major basins (e.g., the Mississippi and Amazon) and very arid regions (e.g., the Taklamakan and central Australia). SHIFT-GAR pairs are the most consistent in mountainous regions (e.g., the Rockies and Andes), aggregated deltas (e.g., eastern Australia and southern Africa), islands (e.g., Indonesia), and inland river basins (e.g., the Tibetan Plateau) where few rivers meet the 5000 km² drainage area threshold of JRC. In contrast, cases where UP pairs align best are less common. UP aligns better with GAR due to their shared large prediction extents, such as around the Caspian Sea. In rare instances where UP-JRC pairs perform best, it is typically in deltas or regions where SHIFT-GAR performs well, such as deltas and islands. This is likely because our method balances consistency between the datasets, but GAR's wider prediction coverage makes this strategy less effective in these infrequent cases."

• **Fig 7** - The color combinations for SHIFT + GAR and JRC + GFPLAIN are indistinguishable.

> **Reply**: Thank you for your advice and we've changed the color combinations accordingly. Please see our revised Fig.8.

General: The authors argue that locally optimized FHG parameters better represent the climatic heterogeneity of the world's basins than using global parameters. I would be more persuaded by a direct comparison of the two methods. That can be

accomplished either by comparing SHIFT to the results of the author's methods but with global FHG parameters (e.g., Fig 6 using $b$ = 0.3, $a$ = 0.01 globally) or a direct comparison of SHIFT and GFPlain to reference data (e.g., Fig 7 without the JRC & GAR and SHIFT & GFPlain250 combinations)

**Reply**: We have performed the analysis as suggested, and please see our reply above and the revision in Section 3.3 that addresses this comment.

**Revision:** Please see the above.

Essentially, the question is: Do locally optimized FHG parameters meaningfully improve the delineation of floodplains over global parameters and is there a spatial pattern of where those improvements occur? Any answer to those questions would be useful information for the community.

**Reply**: Thanks for highlighting again the key contributions of this study (scientifically in addition to contributing to data), that we should better discuss how locally optimized FHG parameters can help better delineate floodplains. We have revised our main texts and figures with additional analyses, attempting to more objectively document the pros and cons of SHIFT.

**Revision:** Please see the above.


**Reviewer #2**

**This manuscript tackles the floodplain mapping at the global scale.** The authors use a methodology to estimate floodplains using a geomorphic approach (integrating heterogeneity), based on past studies such as Nardi et al. (2019). This methodology involves applying a geomorphic descriptor such as HAND (Height Above the Nearest Drainage) and globally optimizing its parameters to delimit floodplains, resulting in a global map with a resolution of 250m. In this study, the authors take a further step by optimizing the parameters of the same geomorphic descriptor (HAND) for more than 200 basins (delimitation at level 3 with respect to HydroBASINS). They consider heterogeneity in the production of a new map on a global scale, which is provided with resolutions of approximately 90m and 1km.

For the calibration and validation of the applied methodology, they relied on 500-year return period maps (JRC, GAR flood maps) and on the 250m Nardi resolution map (GFPlain250m), which presented a general precision greater than 0.85. Additionally, they facilitate access to the results through the following links: available at https://zenodo.org/records/10440609 and the main code at https://github.com/Mostaaaaa/SHIFT_floodplain.

The study is of interest and may be worthy to be published, but some effort should be made to better emphasize the impact of the study. In the following, you will find my comments.

**Reply**: Thanks for your comment and positive evaluation of our work. You'll find our point-to-point replies below.

**Major comments**

• The scaling of hydraulic depth is investigated at the global level, obtaining a very scattered graph. Data seem to be better aligned for larger basins, but some additional effort should be spent to explain the variability observed in other river basins. Climate cannot be the only variable controlling the scaling exponent. Other factors such as rainfall, river morphology, or land use could also impact the result.

> **Reply**: Thank you for raising this reasonable concern, and we fully agree with your interpretation. In this revision, we have expanded our discussion on our analyses of possible relevant factors affecting the empirical parameters. We've approached this expansion in the following ways:

1. **Hypothesis Clarification**: We have better articulated our hypothesis. We consider aridity to be the primary factor influencing the spatial variability of b due to the assumption that in a humid basin, rivers with larger upstream drainage areas (UPA) would have greater dominance over smaller river segments in shaping riverine floodplains. Vegetation should also be related since it is involved in the runoff generation process as well as modulating soil erosion that can be key to floodplain formation. Additionally, other factors such as terrain and soil composition in riverine areas might also influence the results, although the underlying mechanisms are not as intuitive. **It is important to clarify that in no way do we expect perfect relationships between these factors and our derived exponent b, because of the data uncertainties as well as the complex physical interpretations with b.** Based on this hypothesis, we have thoroughly tested the correlation with other possible factors, including Leaf Area Index (LAI), mean elevation, elevation standard deviation, and three types of soil components (clay, silt, and sand). Results indicate that only the Aridity Index exhibits significant correlations with our estimated parameters, while LAI shows significant correlations in the largest basins, and other factors do not present a uniform result. We have included the Aridity Index along with LAI in the revised manuscript as they are results of our primary hypothesis. Other results are provided in the Supplementary Materials. Please refer to the revised Figure 4 and Supplementary Table 1 for these results.

2. **Scale Testing**: We further tested our hypothesis on different scales. We estimated parameters on Level-4 and Level-5 basins to observe if the correlation changes or if any new patterns emerge. Results show that both the Aridity Index and LAI exhibit significant correlations, and terrain factors also show positive but weak correlations, suggesting that these factors might be influential, though the underlying mechanisms could be more complex. See Supplementary Table 1 for the results.

3. **Discussion of Variability**: We have discussed why other basins do not exhibit as strong correlations as the largest basins in the revised Section 4.4. The largest basins are hydrologically connected and thus are expected to have more internally consistent hydrological characteristics from upstream to downstream. In contrast, many other basins are aggregates of smaller basins, so the relationship between the basin-average estimate of parameter 'b' and the Aridity Index might be affected by this aggregation process.

Overall, the comments helped us to perform some meaningful analyses. The insights were included in our revised manuscript and this response letter.

**Revision:**

(Line 662, Section 4.1) "Second, our estimated parameters aim to capture fingerprints from spatially varying hydrological and geomorphic processes that can influence the floodplain extent. We consider aridity as the primary factor influencing the spatial variability of $b$, based on the assumption that in humid basins, rivers with larger upstream drainage areas exert greater dominance over smaller segments in shaping floodplains. Vegetation also plays a role, as it influences runoff generation and modulates soil erosion, both key to floodplain formation. Additionally, factors such as terrain and soil composition might influence the results. Given the data uncertainties and the complex physical interpretations of $b$, it is important to note that we do not expect perfect relationships between these factors and the derived exponent $b$. The correlation analysis indeed aligns with our expectations: AI is statistically significant in explaining the spatial variability of $b$, while LAI plays a role and terrain does not show strong correlations with $b$. Soil compositions (Poggio et al., 2021) do not exhibit a consistent pattern across analyses done at different scales (**Table S1**). Despite the not-so-strong correlation with AI and LAI, its statistical significance supports the effectiveness of our proposed methods, which helps to derive spatially-varying parameters that are also physically meaningful. The parameter $a$ could also encapsulate influences from relevant processes, but its physical interpretation is highly dependent on $b$, as its unit is less uniform (Nardi et al., 2006). Therefore, clarifying the influencing processes of $a$ is beyond the scope of this study."

(Line 751, Section 4.4) "The spatially varying parameters for SHIFT are derived at the scale of HydroBASINS Level-3 basins, which depicts 269 river basins globally with some containing aggregations of smaller basins. These aggregated basins are not hydrologically connected and are less suitable for our thresholding scheme that estimates one set of parameters for each basin, compared to the largest basins which shares internally consistency hydrogeomorphic processes. A possible strategy to improve the scheme is to further divide these basins into smaller sub-basins, but smaller-scale analysis can increase the impact of reference data uncertainties especially in delta regions with high floodplain discordance (**Fig. 5a**). Parameters for Level-4 and Level-5 basins were also calculated (statistics are given in **Table S1**), but many basins had insufficient reference grids to give reliable estimations. Considering the high data noise that

may limit further integration of sub-basin level heterogeneity in estimating parameters, the spatial disaggregation scheme used by SHIFT (i.e., level-03) is sufficient in improving heterogeneity while offering reasonable physical interpretations of the parameters."

**Table 1 (Supplementary Table 1 in the revised manuscript). Correlation of FHG parameter *b* from HydroBASINS Level-3 to Level-5 basins and relevant hydroclimatic factors.** This table presents the correlation of the FHG parameter *b* from HydroBASINS Level-3 to Level-5 basins with relevant hydroclimatic factors. Tests at different scales (Level-4 and Level-5 basins) are added to increase the sample size and confirmed that AI and LAI have statistically significant relationships with the exponent *b*. Terrain factors, specifically elevation mean and standard deviation, exhibited significant but weaker positive correlations with Level-4 and Level-5 basins. Soil factors showed inconsistent and generally insignificant correlations. The results for Level-4 and Level-5 basins were filtered to include only basins with at least 1000 reference grids at a 1-km resolution, ensuring reliable estimation of *b*. The 33 largest basins are those presented in Figure 4 of the revised manuscript. Terrain data are sourced from MERIT-Hydro. Soil data are derived from the Soilgrids 2.0 dataset (Poggio et al., 2021), with zonal averages calculated within a 10-km buffer based on hydrological distance. While some correlations are not very strong, the results meet expectations, highlighting AI as a primary factor, with LAI playing a secondary role, and the other factors showing less observable mechanisms.

| | | Aridity Index | LAI | Elevation Mean | Elevation STD | Clay | Silt | Sand |
|---|---|---|---|---|---|---|---|---|
| Level-3 | All | 0.335*** | 0.083 | -0.007 | 0.121 | 0.152* | 0.170* | -0.041 |
| | Largest | 0.680*** | 0.668*** | -0.165 | 0.208 | 0.314 | -0.134 | -0.042 |
| Level-4 | | 0.338*** | 0.256*** | 0.131** | 0.246*** | -0.067 | 0.050 | -0.003 |
| Level-5 | | 0.405*** | 0.349*** | 0.104*** | 0.188*** | -0.033 | -0.019 | 0.033 |

Note: *** indicates $p < 0.001$, ** indicates $p < 0.01$, * indicates $p < 0.05$.

- In Section 4.1, The authors discuss the uncertainty associated to the parameter b. In this section, results are not clear or do not display a clear pattern. It is also surprising that the results obtained over the larger basins still have a large uncertainty even if the regression function works better.

Reply: Thank you for your comment. We have conducted a deeper investigation into the uncertainty related to parameter 'b', which has led us to restructure and clarify our discussion on this topic. Additionally, we have refined some technical details in the parameter 'b' estimation, resulting in generally smaller uncertainties. Our revisions are as follows:

Conceptually, we've better defined our metric and clarified its interpretation. The metric is now defined as residual uncertainty, which calculates the remaining uncertainty after our data filtering scheme. Physically, this metric assesses how well the data conforms to the power law: a better-conforming set of data result in a narrower range of the estimated $b$ sequence and, consequently, lower standard deviation. By calculating the metric, we aim to see 1) how well the filtered data in these basins align with power law, and 2) the robustness of our parameters, as a lower standard deviation supports the application uniform filtering percentiles globally (see 2.2 Methods). Our goal in refining this section is to provide a clearer explanation of the uncertainty we are addressing and to give a quantitative assessment of our approach to managing it.

Technically, we have modified the technical details of parameter 'b' estimation, specifically the binning parameter in the estimated 'b' sequence generation. The binning parameter determines the number of bins when grouping 'b' values. It should be set higher for samples with high data noise to better filter out outliers. However, setting the binning parameter too high reduces the amount of data available in each bin, and could interfere with the results with few reference grids available. We have improved this process by adding a constraining mechanism to maintain a baseline level of data and clean out empty bins. We tested how the number of bins influences our estimated results. Sensitivity tests show that when the binning parameter exceeds a certain value (150), the estimated 'b' sequence becomes more statistically stable (median, quantiles, and standard deviation). This parameter, which we previously thought was insignificant, has shown that it does not interfere with the overall 'b' values but narrows down the quantiles for large basins with numerous reference points (and subsequently more noise points, e.g., the Mississippi and the Amazon), leading to stabler estimates for these basins. As you pointed out, some large basins (e.g., the Amazon and the

Mississippi) showed greater uncertainties in the previous version of our manuscript. It was possibly due to our prior setting of the binning parameter. Therefore, we've updated the process with the newly set binning parameter and the constraining mechanism, which helps us to rule out another possible source of data noise. Global residual uncertainty now shows a clearer pattern (see revised Fig. 9).

Based on these improvements, we have revised the original Section 4.1 on uncertainty. Please refer to Section 4.2 for the updated content, and Section 2.2 for the improvements above on parameter 'b' estimation.

**Revision:**

(Line 691, Section 4.2) "We also recognize several uncertainties associated with the FHG relation. The primary source of uncertainty comes from the inconsistency between the two reference hydrodynamic datasets across regions, which can be traced back to their model chain errors. Several measures are taken to mitigate the potential influence: we take the intersection of the two datasets as the reference, apply an iterative moving-window scheme to filter the data, and force scaling-law relationships to estimate the parameter b. However, residual uncertainties may still exist due to three aspects: (1) Inconsistencies in terrain data, as both JRC and GAR use SRTM as the inputs while we use MERIT-Hydro; (2) Potential intra-basin heterogeneity of scaling relationships which may lead to unstable estimates; (3) The lack of reference data in certain basins, which lowers our credibility in the estimated parameters. To evaluate how the residual uncertainty influences our FHG parameter estimation, we quantify the uncertainty of b by calculating the standard deviation among all possible b values derived at different percentiles. This metric assesses how well the data conforms to the power law: a better-conforming set of data result in a narrower range of the estimated b sequence and, consequently, lower standard deviation. A lower standard deviation also supports the application uniform filtering percentiles globally (see Methods) and proves robustness of our approach.

Figure 9 reveals the residual uncertainty in parameter b, which ranges from 0 to 0.03 with a median of 0.01. This is considered reasonable for a global median b of 0.3. The pattern is similar to that parameter b itself (Fig. 3), with lower uncertainties in large humid basins (blue color), and greatest uncertainty (red color) observed in arid regions (e.g., The Saharan Regions and western-central Austrailia), mountainous areas (e.g., the Rocky Mountains and the Andes), and deltas (e.g., The Jiaodong Peninsula, the western Mississippi Delta and the Nile

Delta). High residual uncertainty in these regions are possibly due to the particularly strong differences between the reference datasets. For deltas, the great inconsistencies in spatial extents are amplified by their different definition of rivers, as JRC and GAR respectively takes up a stream threshold of 5000 km2 and 1000 km2. It also explains the unexpectedly low b values in deltas observed in Fig. 3. In contrast, the Arctic exhibits generally low uncertainty, likely because only one reference dataset is available above 60□N, reducing discrepancies and thus lowering remaining uncertainty."

(Line 350, Section 2.2) "The binning parameter is tuned to effectively reduce data noise for all basins."

[Figure]

**Fig. 1 (Fig. 9 in the revised manuscript). Spatial pattern of the residual uncertainty of parameter *b* by basins. Residual** uncertainty is quantified as the standard deviation among all possible *b* values derived at different percentiles (see Section 2.2 for details).

- Results should be better described. For instance, it would be valuable to have floodplain patterns obtained from SHIFT with the river network layer and one image showing the differences between SHIFT and a reference map. Additionally, it would be good to enlarge the images in Figure 4.

Reply: Thank you for your advice on better presenting our results. Regarding figures, we've changed our figures on regional differences to better represent regional differences (see our changes in the revised Figure 5).

Revision:

[Figure]

**Fig. 2 (Fig. 5 in the revised manuscript). Geomorphic floodplain extent in SHIFT**. a) Global spatial distribution of floodplains, with major river basins or plains marked out. b) and c) show two cases of that compares SHIFT with GFPlain250m, with background image from © Google Earth on EPSG: 3857 projection. b) locates in the humid Indian-Ganges River basin, while c) locates in the semi-arid yellow river basin in inner Mongolia, China. Major rivers of the region is marked on the map. SHIFT delineates fewer areas in the upstream Ganges River (b) and reduces the floodplain extent outside the Yellow River mainstream (c). It also offers more comprehensive coverage, including the Indus River basin (b) and the Hetao basin (c).

**Minor edits:**

1.  Line 120-123: 'overestimated floodplains in arid or semi-arid area as reported by existing assessments of geomorphic floodplains' (Dhote et al., 2023; Lindersson et al., 2021). In these references, only Lindersson et al. refers to arid areas and their

difficulties. While Dhote et al. only highlights the overestimation and underestimation of the descriptors HAND and TWI respectively, but does not talk about the relationship with arid areas.

**Reply**: Thanks for identifying our negligence and we've removed Dhote et al. in our revised manuscript.

**Revision**:

(Line 125, Section 1) "for example, overestimated floodplains in arid or semi-arid area as reported by existing assessments of geomorphic floodplains (Lindersson et al., 2021)."

2. Line 314-315: 'This iterative process stops either when every data point fits within all moving windows, or if the procedure fails to converge towards a stable solution'. It could explain what is meant by a stable solution, for a better understanding.

**Reply**: "Fails to converge towards a stable solution" refers to the situation where, when dealing with highly noisy or unevenly distributed data, the iterative process fails to reach a stable state within a finite number of steps, resulting in extensive data filtering. The ideal iterative denoising process should filter out fewer and fewer points each time, eventually keeping all points within the 3-sigma range of the sliding window. This assumes that the data is primarily composed of a majority of valid data fitting an assumed overall distribution, combined with a small amount of noise. Under such ideal circumstance, the final sliding window's mean and STD should be an unbiased representation of the population. However, if the valid data is scarce and noise is abundant, it may lead to high natural variability of the data, or the data distribution may be significantly skewed or non-normal, preventing the sliding window's mean and STD from representing the main population, thus leading to a failure of convergence. In this study, non-convergence occurred in very few watersheds with limited reference data points, to address which we established this termination condition. Consequently, we believe that the parameters fitted in these highly noisy watersheds may come with uncertainty. This is further discussed in the revised Section 4.2. We've added a short explanation in the corresponding paragraph to explain it further, but since it's not a major concern we didn't expand in the method section.

**Revision**:

(Line 346, Section 2.2) "This iterative process stops either when every data point fits within all moving windows, or if the procedure fails to converge towards a stable solution (e.g., for highly noised or significantly non-normal data)."

3. Line 343: change 'as the as the' for 'as the'.

   **Reply**: Thanks for pointing out. We've removed the typo.

4. Line 385: only the range of values obtained for the coefficient 'a', For what reason is not presented a graph as in fig.3 of parameter 'b'? If it is possible to provide the values of both parameters, so that this method can be studied at smaller scales focusing future studies in a single basin or a single region and its sub-basins. It would be ideal to base the importance given to the parameter 'b' on the parameter 'a'.

   **Reply**: Thank you for your constructive comment. We do consider parameter 'b' to be more important in our study. In the broader sense of hydraulic geometry, while the value of 'a' is also a determining factor in the floodplain delineation process, the physical interpretation and research focus have historically been more concentrated on the parameter 'b' (since Leopold, 1953). In most cases, the understanding of parameter 'a' is dependent on 'b', and 'b' could be more clearly interpreted as the sensitivity to scale. Thus, in cases where the actual mechanism of the hydraulic geometry relation is not clear, it would be more common to dig into the possible influencing factors of parameter 'b'. In the context of FHG, previous researchers have also primarily focused on 'b'. For example, in a study by Annis et al. (2019), when evaluating the performance of varying FHG parameters across different stream orders, the emphasis was mainly on 'b'.

   In our study, the two parameters have different impacts on delineating floodplains. Parameter 'b' determines whether the river with larger upstream drainage area dominates the floodplain. The difference in impact between large and small rivers is greater when 'b' takes a larger value. This is related to our assumption in our paper that in humid areas, the difference between small and large rivers would be more significant, leading to a stronger dominance of those with larger upstream drainage areas. In comparison, 'a' lacks a unified unit and a clear physical interpretation, and it is highly dependent on 'b'. To provide better understanding, we have provided the spatial distribution of parameter 'a' (see supplementary Figure 1). Unlike the clearer pattern of 'b' of generally better aligning with the Aridity Index, values of 'a' vary largely. For instance, small 'a' values appear in some of the largest river basins (e.g., the Amazon and Yangtze River Basin), possibly balancing out the influence brought by larger 'b' values. While there are possible physical interpretations, it is challenging to interpret 'a' when the underlying mechanism related to 'b' is not clear. Our discussion on the interpretation of parameters was further added to Section 4.1 in Discussion.

Thank you for suggesting that we provide parameters for all basins. We have uploaded all the parameters, along with our confidence levels of all the parameters, in a shapefile our Zenodo repository. We will provide a link in our revised manuscript.

**Revision**:

(Line 676, Section 4.1) "The parameter *a* could also encapsulate influences from relevant processes, but its physical interpretation is highly dependent on *b*, as its unit is less uniform (Nardi et al., 2006). Therefore, clarifying the influencing processes of *a* is beyond the scope of this study."

1. 3b: include a legend.

   **Reply**: Thanks for pointing out. The original Figure 3b was revised to Figure 4 now to expand our discussion on relevant factors. In each of the sub-graphs we've added a legend correspondingly.

2. 3a: change 'estimatio' for 'estimation' in the description.

   **Reply**: Thanks for pointing out. We've removed the typo.


**Reply**: Thank you for your comment on the possible confusion of our conceptualization. We would like to clarify that we are not using hydrodynamic models to create a hydrogeomorphic model but to better determine the floodplain boundary. Although the conceptual definition of a geomorphic floodplain does not involve such a boundary, in practice, it is essential that we obtain some sort of information, be it from hydrodynamic maps or in-situ measurements as the reviewer mentioned in the other comment, to help us define this boundary. By incorporating outputs from hydrodynamic maps (not models), we are obtaining Floodplain Hydraulic Geometry (FHG) parameters for the already established hydrogeomorphic modelling framework, and the maps from hydrodynamic models have proven to be of use. The contribution in our study to the

framework is to add information on spatial variability in parameters.

Conceptually, we believe that different definitions of floodplain boundaries are complementary rather than contradictory, each highlighting different facets of floodplain dynamics. In our case, the concept of a geomorphic floodplain emphasizes the formation process of floodplains, but it is also predominantly shaped by low-probability, high-impact flood occurrences (Lindersson et al., 2021). Considering that FHG describes the extent of inundation depth (hydrological factor) with drainage area (geomorphic factor), our goal of delineating a geomorphic floodplain is subsequently connected with identifying a boundary that encompasses all potentially inundated areas under extreme conditions. **Therefore, we've used two 500-year return period flood inundation maps as references for estimating our parameters, only to ensure a sufficiently large boundary for the carrying out of this algorithm**. **This way, we believe that the geomorphic definition of a floodplain is still obeyed.** While the FHG parameters can be approximated for various return periods (Nardi et al., 2006) and can subsequently be viewed from an inundation perspective, our approach does not focus on a specific return period for inundation. In other words, our goal is not to provide a mere substitute for inundation maps; rather, we aim to consider both the stream's geographical characteristics and hydrological extreme conditions, to identify scaling relationships that align with geomorphic principles, and to offer a more comprehensive understanding of floodplain dynamics.

Thank you again for pointing out this potential confusion on our conceptualization. To better address your concerns, we have largely revised the above discussion on floodplain boundary definition and delineation in the revised Section 4.3 in Discussion. Additionally, to facilitate future studies and reduce computational efforts, we will provide our spatially varying parameters for easier application. These parameters are now available at the same Zenodo repository at https://zenodo.org/records/118358133.

**Revision**:

(Line 728, Section 4.3) "We also dedicate some discussions to the definition of floodplains here as numerous definitions exist for different intended uses. Geomorphically, a floodplain is an accumulation plain along a watercourse, formed by unconsolidated sediment transported and deposited by the stream, usually flooded during high flows (Brierley and Fryirs, 2013). This definition emphasizes the formation process. From a hydrologist or a flood manager's perspective, the floodplain is often associated with inundation attached to certain flood strengths (Krizek et al., 2006), which can be also referred to as the hydraulic floodplain. Alternatively, focusing on material flux exchanges yields different boundaries (Wohl, 2021). We consider these

perspectives not contradictory but complementary in floodplain mapping processes as they highlight different aspects of floodplains. Specifically, geomorphic floodplains are predominantly shaped by low-probability but high-impact flood occurrences (Lindersson et al., 2021), which subsequently connects our goal of delineating a geomorphic floodplain with identifying a boundary that encompasses all potentially inundated areas under extreme conditions. Therefore, we have used two 500-year return period flood maps as references for estimating our parameters, only to ensure a sufficiently large boundary for the carrying out of this algorithm. This way, the geomorphic definition of a floodplain is still obeyed. While the FHG parameters can be approximated for various return periods (Nardi et al., 2006), our approach does not focus on nor involve a specific return period for inundation. In other words, our goal is not to provide a mere substitute for inundation maps. Instead, we aim to leverage a river's geographical characteristics and hydrological extreme conditions, to identify scaling relationships that align with geomorphic principles, and to offer a more comprehensive understanding of global floodplain extents."

(2) Alongside the conceptual limitation, the work is self-contradictory. The authors on and on tag their approach as parsimonious and existing hydrodynamic models as uncertain (see Lines 84-86). Parameterizing HAND with two hydrodynamic model-based flood maps, as the authors did, is in no way a parsimonious method. This is also not a practical method. Because if I don't have hydrodynamic models existing in my area of interest (let's forget about uncertainty for the sake of discussion), I won't be able to reproduce the authors' method.

**Reply**: We respectfully disagree with this point and would like to emphasize that our approach remains parsimonious. Strictly speaking, we did not use hydrodynamic models but rather publicly available flood maps as references, despite their uncertainties and inconsistencies. Thus, no complex models or simulations are involved in our method, as the core process is described by a power-law. The most intricate part of our study is the data filtering scheme, but it still demands significantly less computational effort compared to hydrodynamic models. Besides, we believe that the issue of parsimony and practicality can be better addressed by providing our optimized scaling law parameters. For anyone who wish to reproduce the method/results using terrain data, it is easy to grab our results and derive new maps of their own, thus replicating our method should be feasible.

In addition to the above, in this revision, we have carefully conducted additional investigations into the concern on the uncertainty related to using hydrodynamic maps. Despite their acknowledged inconsistencies, the reference maps we used are informed

by climatic forcing and are subsequently expected to offer a more spatially heterogeneous basis than universal geomorphic parameters. In other words, while we do acknowledge these maps can be uncertain, they contain useful information that can be applied to constrain geomorphic floodplain boundaries. We have thus introduced a rigorous data filtering process to optimize the parameters best conforming to the power law contained within the data. Our results show that the filtered data conform well to the power law (see revised Figure 9), supporting the validity of our approach.

We hope this resolves the "self-contradictory" concern for our work. Revisions have also been made more clearly to address your conceptual concerns: for detailed explanations of using these maps as references, please see the newly added Section 4.1, and for the remaining uncertainty please refer to Section 4.2. Our supplied parameter maps can be found in the zenodo repository for more expert users.

**Revision**:

[revised manuscript text omitted]

Many examples of HAND's parsimonious applications already exist in literature. HAND is parsimonious in operationalized flood prediction systems where a streamflow or stage height (the H in authors' scaling equation) comes from an operational watershed hydrology simulation model followed by a process of automatic synthetic rating curve generation. See examples like https://doi.org/10.31223/osf.io/hqpzg

**Reply**: We thank the reviewer for this comment, and we are actually aware of the alternative thresholding methods for HAND that are available and widely utilized. The

paper you provided outlines two approaches: 1) directly estimating stage height, which is useful when in-situ measurements are available, and 2) using a synthetic rating curve, as also calculated from terrain-based methods. The latter method is indeed effective for large-scale applications and is used by the US National Water Model, but it introduces additional sources of uncertainty as it requires estimated Manning's coefficients for water-stage estimation and it is also computationally very demanding as it has not been accomplished worldwide. Therefore, outside of the United States where high-quality data is available, replicating this globally poses significant challenges.

To compare with, the FHG method requires only terrain input, which is recognized as the least uncertain component in global floodplain mapping when using hydrodynamic models. The necessary information is encapsulated in the parameters, making it easier to identify the influence of each parameter. Therefore, we consider the FHG thresholding approach to be more globally consistent and easily applicable, and still a useful contribution to the community. We have added an additional section on FHG in Section 4.1 and included a paragraph on other thresholding schemes for HAND to address your concerns. We have also supplied parameters for use by future researchers to make it parsimonious.

**Revision**:

(Line 680, Section 4.1) "Third, although alternative thresholding methods that use river discharge and synthetic rating curves exist (e.g., those used by the US National Water Model, Zheng et al., 2018), these methods come with more sources of uncertainty by requiring high-quality data inputs (e.g., gauged discharge, Manning's coefficient). Thus, while they may work well with in-situ observations, replicating this globally poses challenges and is conceptually different from our approach. Our proposed FHG method requires only terrain input, which is recognized as the least uncertain component in global floodplain mapping method (Bates, 2023). By providing the optimized parameters derived here, we consider the FHG thresholding as more globally consistent and easily applicable."

(3) The aridity came out of nowhere. I think bringing aridity into the mix was arbitrary and unnecessary.

**Reply**: Thank you for pointing out this potential confusion, which was also brought up by other reviewers and which we have carefully addressed in this revision. We'd like to clarify that including aridity in our analysis was purposeful and based on our hypothesis. Our estimated parameters aim to capture spatial heterogeneity of

geomorphic floodplain forming factors and, if possible, we should be able to identify significant relationship between examined factors and our derived parameters. Due to uncertainties with the data as well as the scaling law itself, we do not expect the relationship to be perfect. We hypothesized that in humid areas, the stronger discrepancy between small and large rivers would lead to a stronger dominance by larger rivers. While the correlation with the Aridity Index (AI) is not strong, its significance supports our parameter estimation efforts. The largest basins, being hydrologically connected and thus internally consistent in hydrological characteristics, result in stronger correlations with these factors as expected. Despite the seemingly loose correlations, our analyses may still be helpful in identifying geomorphic floodplain-forming mechanisms.

To address your comment regarding the clarity of our purpose, we have clarified our hypothesis and strengthened the tests we conducted in this revision. In terms of writing, we have explained this in the Methods section and elaborated on it in the revised Section 4.1 on our hypothesis with the FHG parameters.

Experimentally, we conducted two additional analyses. First, we included more variables in our analysis, such as LAI, terrain (mean and deviation), and soil factors (soil components in a river buffer). Our hypothesis was that AI would be the most significant factor, with LAI inherently related to AI, while terrain and soil might also be related but with less clear mechanisms. The results showed that AI was indeed the most significant, with LAI only significant in large basins. Other factors exhibit inconsistent correlations with b, also as expected. We also tested the estimation of the parameters at different scales (i.e., Level-4 and Level-5 basins) to increase the sample size. The results showed that AI and LAI have statistically significant relationships with the exponent b, while terrain factors showed significant but much weaker relationships, followed by soil factors that do not show statistically significant relationships with b (see our Supplementary Figure 1, included below).

While the correlations shown in the above analyses may not be very strong, they meet our expectations: AI is significant as the primary factor for explaining the spatial variability of b, LAI plays a role, and terrain might be related but not showing readily detectable correlations with the exponent b.

**Table 1 (Supplementary Table 1 in the revised manuscript). Correlation of FHG parameter *b* from HydroBASINS Level-3 to Level-5 basins and relevant hydroclimatic factors.** This table presents the correlation of the FHG parameter *b*

from HydroBASINS Level-3 to Level-5 basins with relevant hydroclimatic factors. Tests at different scales (Level-4 and Level-5 basins) are added to increase the sample size and confirmed that AI and LAI have statistically significant relationships with the exponent *b*. Terrain factors, specifically elevation mean and standard deviation, exhibited significant but weaker positive correlations with Level-4 and Level-5 basins. Soil factors showed inconsistent and generally insignificant correlations. The results for Level-4 and Level-5 basins were filtered to include only basins with at least 1000 reference grids at a 1-km resolution, ensuring reliable estimation of *b*. The 33 largest basins are those presented in Figure 4 of the revised manuscript. Terrain data are sourced from MERIT-Hydro. Soil data are derived from the Soilgrids 2.0 dataset (Poggio et al., 2021), with zonal averages calculated within a 10-km buffer based on hydrological distance. While some correlations are not very strong, the results meet expectations, highlighting AI as a primary factor, with LAI playing a secondary role, and the other factors showing less observable mechanisms.

| | | Aridity Index | LAI | Elevation Mean | Elevation STD | Clay | Silt | Sand |
|---|---|---|---|---|---|---|---|---|
| Level-3 | All | 0.335*** | 0.083 | -0.007 | 0.121 | 0.152* | 0.170* | -0.041 |
| | Largest | 0.680*** | 0.668*** | -0.165 | 0.208 | 0.314 | -0.134 | -0.042 |
| Level-4 | | 0.338*** | 0.256*** | 0.131** | 0.246*** | -0.067 | 0.050 | -0.003 |
| Level-5 | | 0.405*** | 0.349*** | 0.104*** | 0.188*** | -0.033 | -0.019 | 0.033 |

Note: *** indicates $p < 0.001$, ** indicates $p < 0.01$, * indicates $p < 0.05$.

We've revised our manuscript accordingly to include both the more clearly stated hypothesis and our interpretations. Please refer to the newly performed analyses in Supplementary Table 1, and more objective statements of our parameters and hypothesis in Section 4.1.

**Revision**:

(Line 680, Section 4.1) "Second, our estimated parameters aim to capture fingerprints from spatially varying hydrological and geomorphic processes that can influence the floodplain extent. We consider aridity as the primary factor influencing the spatial

variability of *b*, based on the assumption that in humid basins, rivers with larger upstream drainage areas exert greater dominance over smaller segments in shaping floodplains. Vegetation also plays a role, as it influences runoff generation and modulates soil erosion, both key to floodplain formation. Additionally, factors such as terrain and soil composition might influence the results. Given the data uncertainties and the complex physical interpretations of *b,* it is important to note that we do not expect perfect relationships between these factors and the derived exponent *b*. The correlation analysis indeed aligns with our expectations: AI is statistically significant in explaining the spatial variability of *b*, while LAI plays a role and terrain does not show strong correlations with *b*. Soil compositions (Poggio et al., 2021) do not exhibit a consistent pattern across analyses done at different scales (**Table S1**). Despite the not-so-strong correlation with AI and LAI, its statistical significance supports the effectiveness of our proposed methods, which helps to derive spatially-varying parameters that are also physically meaningful. The parameter *a* could also encapsulate influences from relevant processes, but its physical interpretation is highly dependent on *b*, as its unit is less uniform (Nardi et al., 2006). Therefore, clarifying the influencing processes of *a* is beyond the scope of this study."